# Environmental stimuli shape microglial plasticity in glioma

**Stefano Garofalo[1], Alessandra Porzia[1], Fabrizio Mainiero[2], Silvia Di Angelantonio[3,4], Barbara Cortese[5], Bernadette Basilico[3], Francesca Pagani[4], Giorgio Cignitti[3], Giuseppina Chece[3], Roberta Maggio[2], Marie-Eve Tremblay[6], Julie Savage[6], Kanchan Bisht[6], Vincenzo Esposito[1,7], Giovanni Bernardini[1,8], Thomas Seyfried[9], Jakub Mieczkowski[10], Karolina Stepniak[10], Bozena Kaminska[10], Angela Santoni[1,8], Cristina Limatola[1,11]\***

[1]IRCCS Neuromed, Pozzilli, Italy; [2]Department of Experimental Medicine, Sapienza University, Rome, Italy; [3]Department of Physiology and Pharmacology, Sapienza University, Rome, Italy; [4]Center for Life Nanoscience, Istituto Italiano di Tecnologia, Rome, Italy; [5]Consiglio Nazionale delle Ricerche, Institute of Nanotechnology, Rome, Italy; [6]Département de médecine moléculaire, Université Laval, Quebec, Canada; [7]Department of Neurology and Psychiatry, Sapienza University, Rome, Italy; [8]Department of Molecular Medicine, Sapienza University, Rome, Italy; [9]Biology Department, Boston College, boston, United States; [10]Neurobiology Center, Nencki Institute of Experimental Biology of the Polish Academy of Sciences, Warsaw, Poland; [11]Department of Physiology and Pharmacology, Sapienza University, Laboratory affiliated to Istituto Pasteur Italia – Fondazione Cenci Bolognetti, Rome, Italy

**\*For correspondence:**
cristina.limatola@uniroma1.it

**Competing interests:** The authors declare that no competing interests exist.

**Abstract** In glioma, microglia and infiltrating macrophages are exposed to factors that force them to produce cytokines and chemokines, which contribute to tumor growth and to maintaining a pro-tumorigenic, immunosuppressed microenvironment. We demonstrate that housing glioma-bearing mice in enriched environment (EE) reverts the immunosuppressive phenotype of infiltrating myeloid cells, by modulating inflammatory gene expression. Under these conditions, the branching and patrolling activity of myeloid cells is increased, and their phagocytic activity is promoted. Modulation of gene expression depends on interferon-(IFN)-γ produced by natural killer (NK) cells. This modulation disappears in mice depleted of NK cells or lacking IFN-γ, and was mimicked by exogenous interleukin-15 (IL-15). Further, we describe a key role for brain-derived neurotrophic factor (BDNF) that is produced in the brain of mice housed in EE, in mediating the expression of IL-15 in CD11b[+] cells. These data define novel mechanisms linking environmental cues to the acquisition of a pro-inflammatory, anti-tumor microenvironment in mouse brain.
DOI: https://doi.org/10.7554/eLife.33415.001

## Introduction

Glioblastoma (GBM) is a grade IV (WHO) brain tumor, characterized by high cell proliferation, active angiogenesis and invasion capability. Tumor cells produce immunosuppressive molecules that, rather than promoting the immune reaction against the tumor, recruit T regulatory cells and glioma-associated myeloid cells (GAMs). The latter population comprises microglia, peripheral monocytes, perivascular macrophages and myeloid-derived suppressor cells (MDSC) (*Quail and Joyce, 2017*), and may represent up to 30% of total tumor mass (*Graeber et al., 2002*; *Markovic et al., 2009*). GAMs play an active part in the enhancement of the invasive and proliferative properties of GBM, further

contributing to the immunosuppressive phenotype of the tumor microenvironment (*Markovic et al., 2005*; *Coniglio and Segall, 2013*). There is a wide debate regarding the definition of GAM phenotype: the initial descriptions identified these cells as M2 type, by analogy to the phenotypic distinction originally used for macrophages (*Sica et al., 2008*; *Murray et al., 2014*). Subsequent studies identified a plethora of phenotypes amongst GAMs, with additional differences also observed between mice and humans (*Szulzewsky et al., 2016*). Murine GAMs express several genes that have anti-inflammatory properties, and many efforts aim to promote the expression of pro-inflammatory genes by GAMs, as this phenotype is associated with better patient prognosis and increased survival in animal models (*Pyonteck et al., 2013*). However, epidemiological studies have demonstrated that factors involved in chronic inflammation may increase the incidence of several tumor types (*Mantovani et al., 2008*).

Environmental stimuli have been shown to shape brain functions, modulating plasticity and learning and memory functions, and reducing the impact of neurodegenerative diseases in animal models and in patients (*Sale et al., 2014*). Brain-derived neurotrophic factor (BDNF) is a key cerebral mediator of these phenomena (*Branchi et al., 2004*). We have previously demonstrated that housing mice in an enriched environment (EE) reduced intracranial glioma growth, with indirect mechanisms acting through innate immune natural killer (NK) cells, and with direct effects of BDNF on the truncated isoform 1 of the tropomyosin receptor kinase B (TrkBT1), which is expressed by tumor cells. Anti-tumor activity of an EE has been shown in different cancers (*Cao et al., 2010*; *Nachat-Kappes et al., 2012*; *Garofalo et al., 2015*), even if some results were not confirmed (*Westwood et al., 2013*). Environmental stimuli also shape immune functions, modulating NK cells (*Garofalo et al., 2015*; *Song et al., 2017*), T helper cells (*Rattazzi et al., 2016*), and myeloid cells (*Chabry et al., 2015*). NK cells exert anti-tumor function by direct cytotoxic activity, producing several cytokines, chemokines and growth factors (*Vivier et al., 2011*; *Guillerey and Smyth, 2016*; *Stabile et al., 2017*). Among the cytokines, interferon (IFN)-γ contributes to tumor immune-surveillance and interacts with T cells, dendritic cells and monocytes (*Kaplan et al., 1998*; *Dunn et al., 2006*). IFN-γ is a pleiotropic cytokine, which has anti-tumor activity in different neoplasms by inhibiting cell proliferation and exerting anti-angiogenic activity (*Parker et al., 2016*). In particular, IFN-γ counters the acquisition of a pro-tumor phenotype by GAMs and reverses the immunosuppressive phenotype (*Duluc et al., 2009*).

The maturation and survival of NK cells require IL-15, which also stimulates IFN-γ production (*Yu et al., 2006*), and mice lacking IL-15 or one of its signaling components are devoid of NK cells (*Koka et al., 2003*).

Despite the advancement of knowledge regarding the interactions between immune cells and the microenvironment of brain tumors, several aspects need further investigation. In this study, we focused on the effect of macro-environment on GAM phenotype. We report that housing mice in EE changes GAM phenotype by reducing the expression of anti-inflammatory genes and increasing the expression of pro-inflammatory ones. Exploring the underlying mechanisms, we identified NK cells as mediators of this immunodeviation, with key intermediate roles played by BDNF, IL-15 and IFN-γ.

## Results

### Housing mice in EE modulates gene expression, K$^+$ channel functions and morphology of GAM

To investigate the effect of the macroenvironment on GAMs, C57BL/6 mice housed in standard environment (SE) or EE for 5 weeks after weaning were injected with syngeneic glioma cells (GL261) into the right striatum. Animals were then returned to their original housing cages. A preliminary set of confirmatory experiments was performed to cross validate our experimental system (see Materials and methods). After 17 days, mice were sacrificed and CD11b$^+$ cells were isolated from the ipsi- (ILH) or contra-lateral (CLH) cerebral hemispheres. RT-PCR analysis revealed that CD11b$^+$ cells from the ILH (compared to those from the CLH) of mice housed in SE (SE mice) preferentially express anti-inflammatory markers that include *Chil3*, *Mrc1*, *Arg1* and *Retnla* (*Gabrusiewicz et al., 2011*). Some pro-inflammatory genes, like *Tnfa* and *Nos2,* were also upregulated in the ILH, whereas no differences were observed for *Cd86* and Il1b. In EE, the gene expression of CD11b$^+$ cells isolated from the ILH was deeply modified, showing the significant increase of pro-inflammatory and

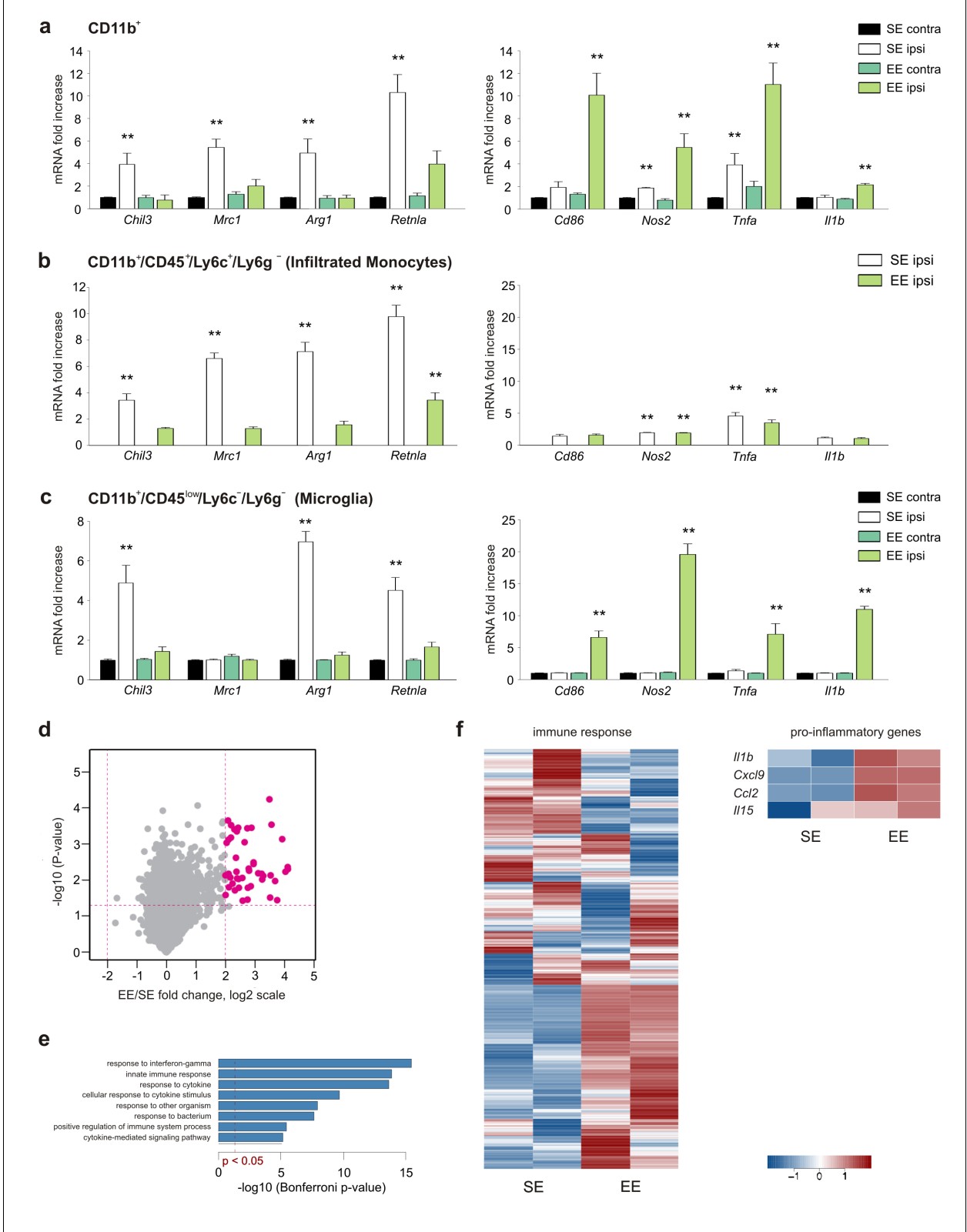

**Figure 1.** EE modulates myeloid cell phenotype. (**a**) RT-PCR of anti- (*Chil3, Mrc1, Arg1, Retnla)* and pro-inflammatory (*Cd86, Nos2, Tnfa, Il1b*) genes in CD11b + cells sorted from ILH and CLH of GL261-bearing mice housed in EE or SE. Data are the mean ± S.E.M., \*\*p<0.01 versus CLH by one-way ANOVA; n = 6. (**b–c**) RT-PCR of anti- (*Chil3, Mrc1, Arg1, Retnla)* and pro-inflammatory (*Cd86, Nos2, Tnfa, Il1b*) genes in CD45[low]/CD11b[+]/Ly6c[−]/Ly6g[−] and CD45[+]/CD11b[+]/Ly6c[+]/Ly6g[−]cells sorted from ILH and CLH of GL261-bearing mice housed in EE or SE. Data are the mean ± S.E.M., \*\*p<0.01

*Figure 1 continued on next page*

*Figure 1 continued*

versus CLH (taken from CD11b + cells in [a]), by one-way ANOVA; n = 5. (**d**) Comparison of gene expression profiles resulting from EE (n = 2) compared to the SE exposure (n = 2) in microglia from ILH. Each point on the scatterplot represents a gene. The X-axis shows expression ratio (EE/SE) on the $\log_2$ scale. Y-axis presents corresponding p-values ($-\log_{10}$[p-value]). Genes that are significantly upregulated after EE exposure (t-test p-value<0.05, fold-change >4) are marked by large purple dots. Global gene expression was determined using Affymetrix microarrays. (**e**) Results of Gene Ontology over-representation analysis (Bonferroni corrected p-values <0.05) of the genes that are upregulated in microglia after EE exposure (**d**). X -axis corresponds to $-\log_{10}$-transformed p-values. (**f**) Expression profile of genes associated with immune response according to the Gene Ontology database and manually selected pro-inflammatory genes.

DOI: https://doi.org/10.7554/eLife.33415.002

The following source data and figure supplement are available for figure 1:

**Source data 1.** Raw data and detailed statistical analysis report.

DOI: https://doi.org/10.7554/eLife.33415.004

**Figure supplement 1.** EE modulates tumor size and myeloid cell phenotype in mice injected with CT-2A cells.

DOI: https://doi.org/10.7554/eLife.33415.003

reduction of anti-inflammatory genes (*Figure 1a*). Similar results were obtained when studying CD11b[+] cells isolated from the brain of mice injected with a different, less immunogenic murine cell line, CT-2a. Also in this condition, tumor size was significantly reduced in EE mice as compared to SE mice (*Figure 1—figure supplement 1a,b*).

To further define the relative contributions of microglia and myeloid cells recruited from the periphery, we isolated CD11b[+]/CD45[low]/Ly6C[−]/Ly6g[−] (corresponding to microglia) and CD11b[+]/CD45[+]/Ly6C[+]/Ly6g[−] (corresponding to infiltrating monocytes) cell populations from the brain hemispheres of tumor-bearing SE or EE mice (*Figure 1b,c*). The results of the gene expression analysis indicate that in both housing conditions, monocytes only infiltrate the ILH and are virtually absent from the CLH (*Figure 1b*), whereas microglia are present in both hemispheres (*Figure 1c*). In the ILH of EE mice, the expression of anti-inflammatory genes was reduced in both microglia and infiltrating monocytes, whereas the expression of anti-tumor, pro-inflammatory genes was increased only in microglia (*Figure 1b,c*). Consistently, when global expression profiles of the same cell populations were analyzed, similar results were obtained. Genes that were upregulated in microglia from the ILH of EE mice were enriched in Gene Ontology (GO) terms such as 'response to interferon-gamma' and 'innate immune response' (*Figure 1d,e*). We also observed the divergent activation of genes that are associated with 'immune response', which was further supported by the differential activation of pro-inflammatory genes (*Figure 1f*).

As potassium channels are differentially expressed in different microglia phenotypes (*Nguyen et al., 2017*), we analyzed their functional expression to further characterize the microglial activation state. Patch-clamp recordings were performed in GFP[+] cells of acute brain slices from glioma bearing *Cx3cr1[+/GFP]* mice, which comprise GAMs, dendritic cells, and NK cells (*Jung et al., 2000*). As shown in *Figure 2a*, only GFP[+] cells in the ILH have outward-rectifying potassium currents ($K_{or}$, flowing through Kv1.3 and Kv1.5), which are absent in the CLH, and the average current amplitude is not modified by exposure to EE. Focusing on the peritumoral region, the occurrence of $K_{or}$ currents is increased by EE (*Figure 2b*). In the CLH, the amplitude of the inward-rectifying K currents ($K_{ir}$, carried by Kv2.1 channels) is increased in the GFP[+] cells of EE mice (*Figure 2c*). According to *Richter et al. (2014)*, and from the passive membrane properties, we identified these cells as microglia (see Materials and methods). We then analyzed GFP[+] cell morphology by two-photon microscopy, measuring cell branching and territory (i.e. mean area covered by single cells). Our data show that, in the peritumoral region of EE mice, GFP[+] cells have an increased number and length of branches, and cover a wider parenchymal region (*Figure 2d*). On the other hand, these cells display a reduced patrolling activity, as indicated by reduced velocity and process extension into the brain parenchyma (*Figure 2e*), which is probably balanced by their wider coverage (*Figure 2a*). We also observed that only GFP[+] cells in the peritumoral area rearrange their processes toward a pipette-guided focal application of ATP. The speed of these movements increases in EE (*Figure 2f*). This behavior could be due to an increased expression of *P2RY12* (*Figure 2g*) in CD11b[+] cells isolated from the brain of EE mice.

Myeloid cell morphology was also investigated inside the tumor. Two-photon analysis shows that in EE mice, GFP[+] cells inside the tumor have increased process length and total surface area

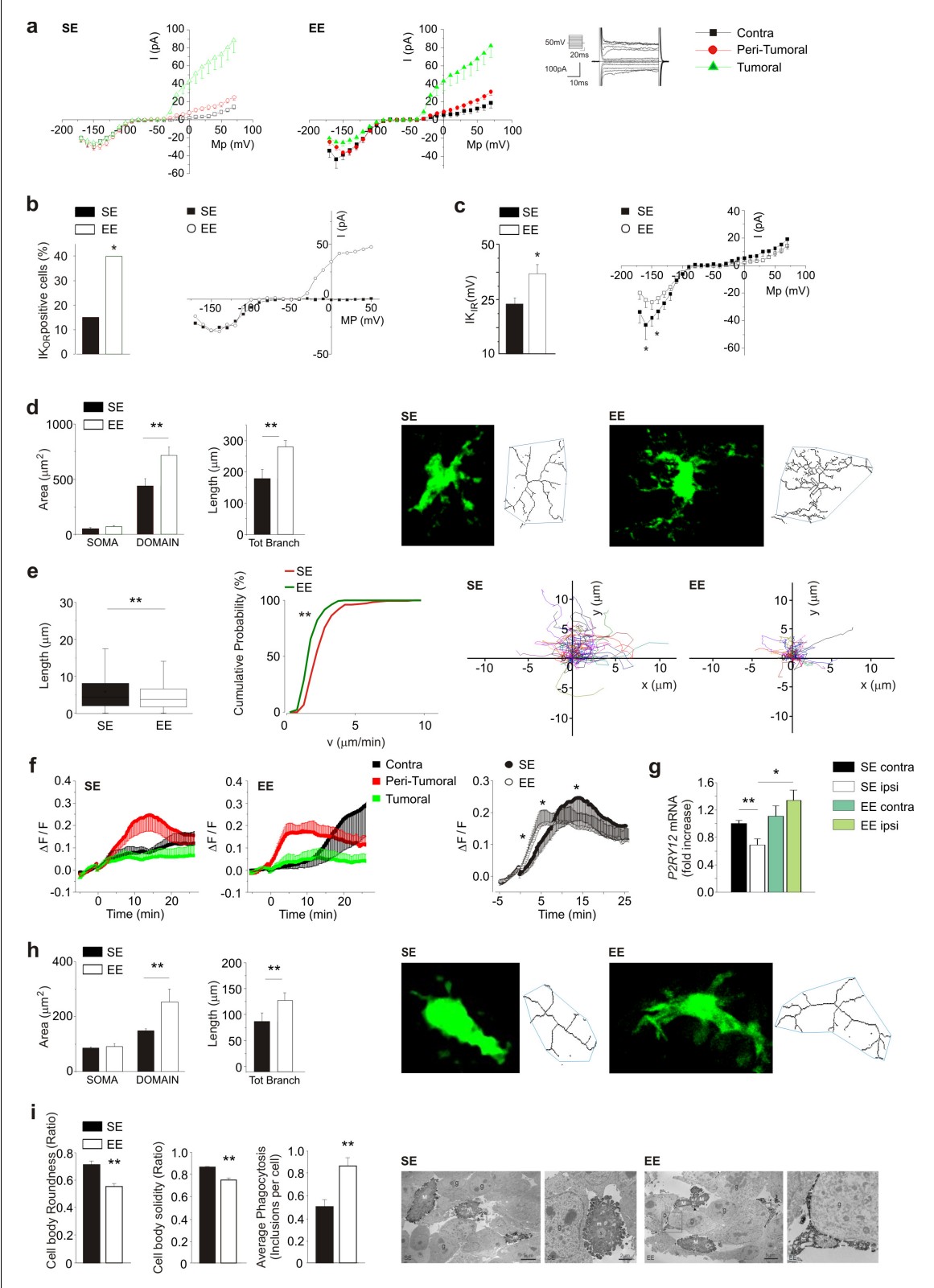

**Figure 2.** Effect of EE on myeloid cell morphology. (a) Left: current/voltage relationship of microglia cells in response to voltage steps stimulation (steps from −170 to +70 mV, only one out of two steps are shown; holding potential −70 mV) in CLH (n = 38/9 mice), peritumoral area (n = 60/9 mice) and inside the tumor (n = 57/9 mice) of SE housed, GL261-bearing mice. Right: Current/voltage relationship of microglia cells in CLH (n = 27/9 mice), peritumoral area (n = 57/9 mice) and inside the tumor (n = 64/9 mice) of EE mice. (b) Percentage of GFP$^+$-cells expressing $K_{or}$ currents in

*Figure 2 continued on next page*

*Figure 2 continued*

the peritumoral area in SE and EE mice (*p<0.05, z-test). Representative current/voltage relationships are shown on the right. (c) Amplitude of $K_{ir}$ current expressed by GFP$^+$ cells in the peritumoral area in SE and EE mice (*p<0.05, z-test). Representative current/voltage relationships are shown on the right. (d) Left: quantification of area of the soma and scanning domain of GFP$^+$ cells measured by ImageJ in slices from GL261-bearing mice housed in SE or EE, as indicated (15 cells, 6 slices, 4 mice per condition, **p=0.0034, *t*-test). Center: total branch length of GFP$^+$ cells in SE and EE (**p=0.0046, *t*-test). Right: representative images of maximum intensity projections of a confocal z-stack imaging on peritumoral area of GFP$^+$ cells in SE and EE mice, converted to binary images and then skeletonized by the Analyze Skeleton plugin in Image J (green lines). (e) Length (left, **p=1.35E-24) and cumulative probability histogram of mean velocity (right, p=0.0053) of spontaneous (basal) movement of all single processes of the peritumoral area measured in SE (red, n = 177 tracks, 6 mice) and in EE (green, n = 210 tracks, 6 mice). Right images show reconstruction of basal process migration tracks in SE and EE. Individual tracks were aligned to the origin. (f) Time course of fluorescent ratio (ΔF/F) measured in a circle (10 μm radius) centered on the tip of the ATP puff pipette, in CLH (black, n = 9/9, SE; n = 7/9, EE), peritumoral (red, n = 11/9, SE; n = 12/9, EE) and intra-tumoral (green, n = 9/9, SE; n = 11/9, EE) areas of slices from *Cx3cr1*$^{+/GFP}$ mice housed in SE or EE. Note that the fluorescence increases around the pipette tip only in the peritumoral area (p<0.05; one-way ANOVA). Right: time course of fluorescence ratio evaluated in the peritumoral area of *Cx3cr1*$^{+/GFP}$ mice housed in SE (red, n = 11) and EE (black, n = 12) (p=0.041 at 3 min; p=0.035 at 8 min; p=0.018 at 15 min; *t*-test). (g) RT-PCR of *P2RY12* gene in CD11b$^+$ cells sorted from ILH and CLH of GL261-bearing mice, housed in SE or EE. Data are the mean ± S.E.M., *p<0.05 **p<0.01 versus CLH by one-way ANOVA, n = 4. (h) Representative SE and EE z-projections of GFP$^+$ cells (skeletonized as above) into the tumoral area of *Cx3cr1*$^{+/GFP}$ mice. Left: 13 cells, 6 slices, 4 mice per condition, **p<0.01, Student's t-test. Note that the scanning domain in the tumoral area was significantly smaller than that in the peritumoral area (p=0.021; *t*-test). Right: bar chart reporting the morphometric analysis of microglia branches (total branch length) of SE and EE microglia p=0.0031, *t*-test). (i) Ultrastructural analysis of cell body circularity and solidity, as well as average number of phagocytic inclusions in SE versus EE myeloid cells. Representative pictures showing IBA1-stained myeloid cells from both conditions are also provided. M = microglial cell body; m = microglial process; s = secretion granule; in = phagocytic inclusion; g = glioma cell. Data are the mean ± S.E.M., **p<0.01 by unpaired *t*-test; n = 25 cells from two animals in SE and n = 45 cells from three animals in EE.

DOI: https://doi.org/10.7554/eLife.33415.005

(*Figure 2h*). Electron microscopy showed ultrastructural differences too: in EE mice, IBA1$^+$ cells inside the tumor have reduced cell body roundness, indicating ramified (versus amoeboid) morphologies, as well as reduced solidity, being delineated by smoother plasma membranes (*Figure 2i*). Indeed, there was an increased prevalence of small 'pseudopodia' protruding from the cell bodies in the SE mice. In EE mice, microglia more often extended long process ramifications, sometimes wrapping around the tumor cells. They also contained an increased number of phagocytic vacuoles, sometimes with cellular elements showing signs of digestion. Secretory granules were observed in both housing conditions. Altogether these data demonstrate that in tumor-bearing EE mice, the phenotype of microglia and infiltrating myeloid cells is changed towards a patrolling, pro-inflammatory state, which actively trys to re-establish brain homeostasis and to restrain tumor growth.

## NK cells mediate the effects of environment

Having previously shown that NK cells actively participate in the process of tumor reduction induced by EE (*Garofalo et al., 2015*), we wondered whether NK cells could contribute to the EE-induced phenotypic switch of CD11b$^+$ cells. With this aim, glioma-bearing mice were treated with an antibody against NK1.1 to deplete the NK cell population. The efficacy and specificity of this treatment has been shown previously (*Garofalo et al., 2015*). In SE, NK cell depletion per se modifies the phenotype of CD11b$^+$ cells in tumor-bearing brains towards an anti-inflammatory phenotype (*Figure 3a*; note that the scale values differ from those in *Figure 1a*).

In EE mice with glioma, NK cell depletion blocks CD11b$^+$ cell polarization towards a pro-inflammatory phenotype, demonstrating the importance of NK-CD11b$^+$ intercellular communication for these effects. To explore the mechanisms underlying NK-CD11b$^+$ cell communication, we first looked at their relative distribution within the tumor mass, using two-photon microscopy. We focused on measuring the distance between NK cells and myeloid cells, using *Cx3cr1*$^{GFP/+}$ mice in which microglia are green, whereas GL261 glioma cells were visualized with RFP, and NK cells were stained with an Alexa-Fluor 633-conjugated NK1.1 Ab. The data indicate that in the EE condition, the number of NK cells that are in contact with myeloid (GFP$^+$) cells is not significantly different from that in the SE condition (*Figure 3—figure supplement 1a*). On the other hand, in EE conditions, the frequency of direct contacts between NK cells and glioma cells was significantly increased (*Figure 3b*), suggesting that these structural interactions underlie the increased NK cell degranulation (see below). Electron microscopic analysis of tumors (*Figure 3c*) further shows that in EE mice, IBA1$^+$ cells with the ultrastructural characteristics of microglia (*Tremblay et al., 2012*) and

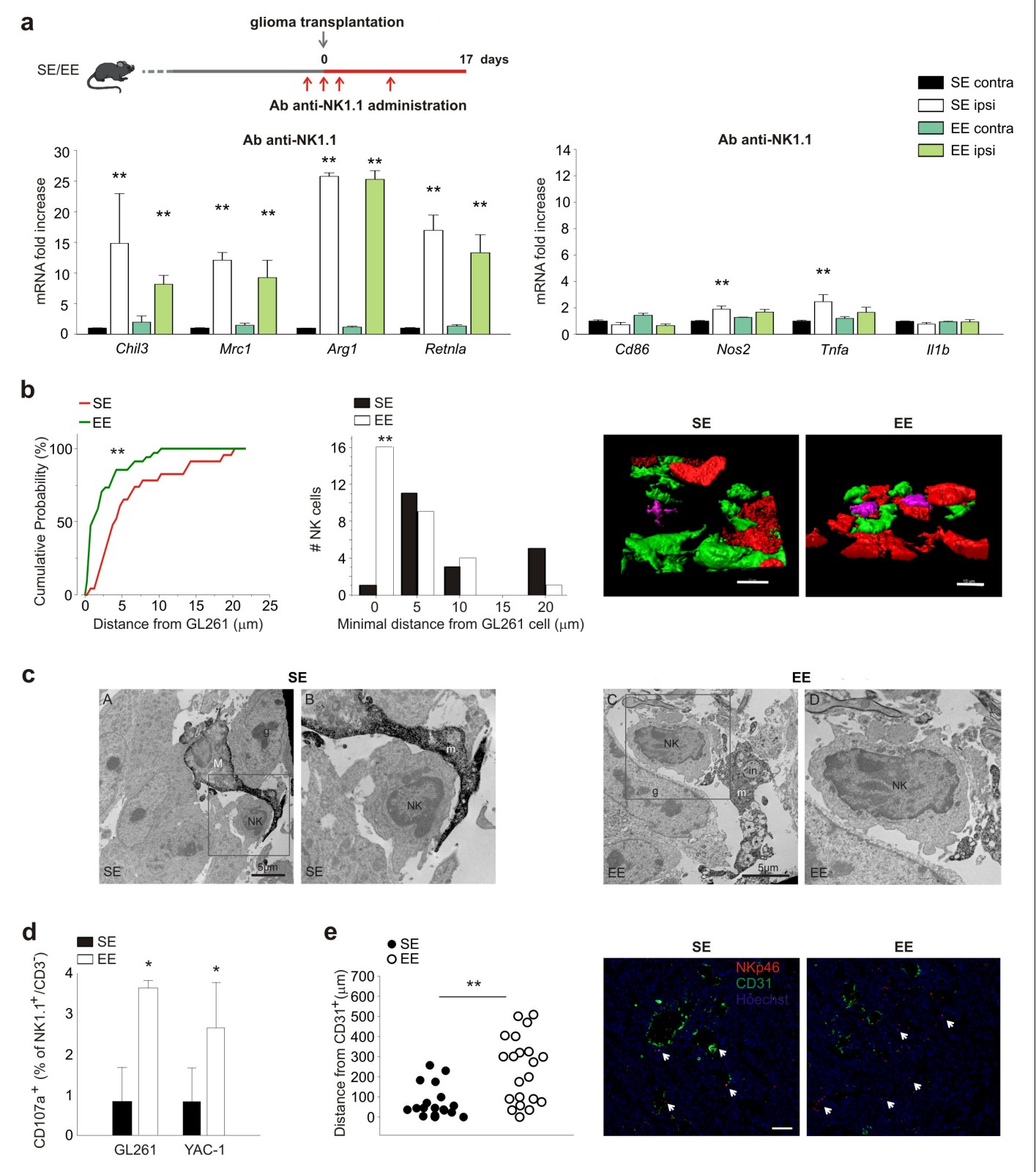

**Figure 3.** NK cells are modulated by EE. (**a**) RT-PCR of anti- (*Chil3*, *Mrc1*, *Arg1*, *Retnla*) and pro-inflammatory (*Cd86*, *Nos2*, *Tnfa*, *Il1b*) genes in CD11b[+] cells sorted from ILH and CLH in vehicle and NK1.1 Ab-treated GL261-bearing mice, housed in SE or EE. Scheme of Ab-NK1.1 administration above. Data are the mean ± S.E.M., **p<0.01 versus CLH by one-way ANOVA; n = 6. (**b**) Cumulative distributions of distances of GL261 3D iso-surface from NK cells iso-surface, in brain slices from SE- and EE-housed mice. Note that in SE-housed mice, NK cells are significantly more distant from glioma cells

*Figure 3 continued on next page*

*Figure 3 continued*

than they are in EE (SE n = 14, 6 slices, 4 mice; EE n = 16, 6 slices/4 mice); **p<0.01, Kolmogorov-Smirnov test). Top: representative 3D reconstruction, by Imaris software, of GFP⁺ cells, Tag-RFP GL261 cells, and NK1.1-positive cells (Alexa Fluor 633 conjugated secondary Ab) in the tumoral area of brain from SE- (left) and EE- (right) housed mice. (Scale bar 10 μm.) (**c**) Ultrastructural evidence of direct contacts between microglia and NK cells inside the tumor. Examples of microglial processes wrapping around NK cells are provided for both SE and EE. In EE, the increased prevalence of phagocytic inclusions containing intact elements (in; myelinated axon being internalized) or other types of debris that are being digested (*) is shown. Filopodia protruding from the NK cell body where it touches a microglial process are observed in the images. M = microglial cell body; m = microglial process; NK = NK cell; in = phagocytic inclusion; g = glioma cell. (**d**) NK cells, isolated from the brains of EE and SE GL261-bearing mice, were incubated with GL261 or YAC-1 cells, and degranulation was assessed by FACS analysis of CD107a⁺ cells. Average values ± SD of CD107a⁺ cell frequency upon GL261 or YAC-1 cell co-incubations, minus blanks (degranulation in the absence of targets, in three independent experiments). Student's *t*-test, *p=0.011. (**e**) Mean distances of NK cells from endothelial cells in vessels (CD31⁺ cells) 17 days after GL261 cell transplantation in SE or EE conditions (n = 4 mice per condition; **p=0.001 Student's *t*-test, scale bar 0.1 mm). Representative immunofluorescence is shown on the right.

DOI: https://doi.org/10.7554/eLife.33415.006

The following figure supplement is available for figure 3:

**Figure supplement 1.** NK cells distance from GFP+ cells and NK cell activation in EE.

DOI: https://doi.org/10.7554/eLife.33415.007

juxtaposed to NK cells contain an increased number of phagocytic vacuoles, sometimes containing cellular elements in the process of being digested. We confirmed (*Garofalo et al., 2015*) that in the brain of EE mice, there is an increased frequency of NK cells expressing granzyme B and IFN-γ (see *Figure 3—figure supplement 1b*), and this leads to an increased degranulation of NK cells ex vivo (not shown) which was further increased when challenged against tumor cells (GL261 and YAC-1 cells, *Figure 3d*).

NK cells were also more distant from CD31⁺ endothelial cells, probably because of an increased colonization of the brain parenchyma in the EE condition (*Figure 3e*).

## IFN-γ released by NK cells plays a critical role in the control of glioma growth

Investigating the potential role of IFN-γ as mediator of NK-cell-mediated effect, we demonstrated that glioma-bearing EE mice express higher levels of IFN-γ in the ILH, and that NK cell depletion abolishes this increase (*Figure 4a*). To confirm the importance of IFN-γ produced upon housing in EE, mice were treated with an IFN-γ—blocking antibody (Ab-anti-IFN-γ), before analyzing the expression of pro- and anti-inflammatory genes in CD11b⁺ cells isolated from their brains. The efficacy of anti-IFN-γAb treatment was verified by analysis of MHCII protein expression in infiltrating CD11b⁺ cells, brain IgG staining (*Figure 4—figure supplement 1a–b*), and CD31⁺ cell density in the tumor area, as described below. IFN-γ depletion completely abolishes the effect of EE on CD11b⁺ cell phenotypic changes (*Figure 4b*). This treatment has consequences similar to those of NK cell depletion (*Garofalo et al., 2015*), reducing the effect of EE on tumor volume and inhibiting the pro-survival effect (*Figure 4c–d*). EE and IFN-γ depletion alone have opposite effects on angiogenesis. EE housing reduces the extent of CD31 staining, probably because of the production of angiostatic chemokines (*Coughlin et al., 1998*); whereas IFN-γ depletion per se increases angiogenesis. When the two treatments are combined, the reduction induced by EE is less pronounced but still maintained (*Figure 4e*). The blunted effect of EE on tumor size reduction was at least partially dependent on a reduced inhibition of tumor cell proliferation, as more Ki67⁺ cells were present in the tumor mass of IFN-γ-depleted mice (*Figure 4f*), in accordance with the anti-proliferative activity of IFN-γ. The effect of IFN-γ depletion was also investigated in mice injected with CT-2a glioma cells: data shown in *Figure 4—figure supplement 2a–b* demonstrate similar effects on tumor size and modulation of CD11b⁺ cell phenotype.

All together, these data indicate that the IFN-γ produced by NK cells upon housing in EE is responsible for the phenotypic modification of CD11b⁺ cells, reduction of tumor size, and increased survival of glioma-bearing mice.

To investigate the effect of IFN-γ on human GAMs, we stimulated human GBM tissues with IFN-γ or vehicle for 24h and then analyzed the CD11b⁺ cell population. *Figure 4g* shows that IFN-γ treatment decreases expression of the human anti-inflammatory genes *Cd163*, *Mmp12* and *Tgfb*, and increases expression of the pro-inflammatory genes *Il12a*, *Nos2* and *Cxcl10*.

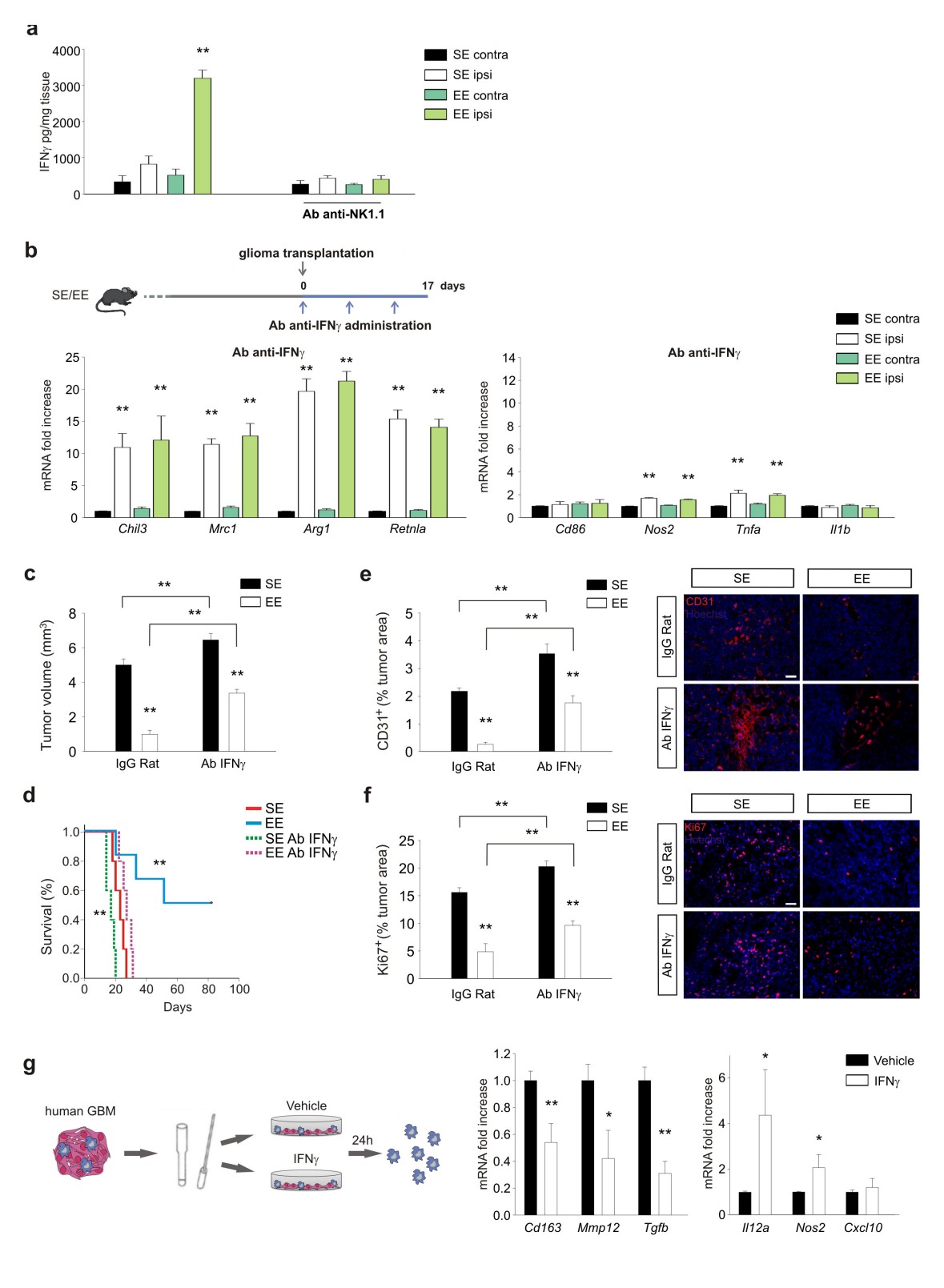

**Figure 4.** IFN-γ mediates the effects of EE on myeloid cells. (a) Expression of IFN-γ in CLH and ILH of GL261-bearing mice treated with vehicle or Ab anti-NK1.1 as indicated, and housed in SE or EE. (n = 5–4, **p<0.01, one-way ANOVA.) (b) RT-PCR of anti- and pro-inflammatory genes in CD11b+ cells sorted from ILH and CLH in vehicle and Ab-anti-IFN-γ GL261-bearing mice, housed in SE or EE. Above: scheme of Ab-anti-IFN-γ administration. Data are the mean ± S.E.M., **p<0.01 versus CLH, one-way ANOVA; n = 4. (c) Analysis of GL261 tumor volume (expressed in mm³ ± s.e.m.) in IgG Rat

*Figure 4 continued on next page*

Figure 4 continued

or Ab-anti-IFN-γ-treated mice, housed in SE or EE; n = 5; **p<0.01, two-way ANOVA). (d) Kaplan–Meier analyses of GL261-transplanted mice exposed to SE or EE upon vehicle or Ab-anti-IFN-γ treatment; n = 5; log-rank test **p<0.01. (e) Quantification of CD31$^+$ cells (mean ± S.E.M. of CD31$^+$area as % of the tumor area, **p<0.05, one-way ANOVA, n = 4 mice per condition) 17 days after GL261 transplantation in mice treated with IgG-Rat or Ab-anti-IFN-γ, as indicated. Representative immunofluorescences are shown on the right. (f) Quantification of Ki67$^+$ cells (mean ± s.e.m. of Ki67$^+$ area as % of the tumor area, **p<0.05, one-way ANOVA, n = 4 mice per condition) 17 days after GL261 transplantation in mice treated with IgG-Rat or Ab-anti-IFN-γ, as indicated. Representative immunofluorescences are shown on the right. (g) RT-PCR of human pro- (Cxcl10, Nos2 and Il12a) and anti-inflammatory (Cd163, Mmp12 and Tgfb) genes in CD11b$^+$ cells sorted from patient-derived GBM tissue, after tissue treatment with IFN-γ (20 ng/ml, 24 hr) or vehicle. Above: scheme of human GBM treatment. Data are the mean ± S.E.M., for Nos2 and Il12a *p=0.029, for Tgfb and mmp12**p=0.002, for cd163 p=<0.001 versus vehicle, Student's t-test, n = 3–6.
DOI: https://doi.org/10.7554/eLife.33415.008

The following figure supplements are available for figure 4:

**Figure supplement 1.** Control experiments to verify the efficacy of Ab-IFN-γ treatment.
DOI: https://doi.org/10.7554/eLife.33415.009
**Figure supplement 2.** Effect of Ab-IFN-γ on tumor size and gene expression of myeloid cells in EE mice injected with CT-2A cells.
DOI: https://doi.org/10.7554/eLife.33415.010

## IL-15 is required as an intermediate modulator of the effect of EE on NK cells

We next sought to verify the hypothesis that IL-15 produced by CD11b$^+$ cell in the brain of EE mice (Garofalo et al., 2015) could be involved in the IFN-γ-mediated effect of NK cells on the polarization of CD11b$^+$ cells. To this aim, IL-15 was infused into mouse brains for one week, starting 10 days after the tumor implantation. Previous experiments demonstrated that this treatment increases the survival of glioma-bearing mice and reduces tumor size (Garofalo et al., 2015). We here confirm that tumor size reduction also occurred in mice injected with CT-2a cells or cancer stem cells (CD133$^+$-GL261 cells) and in SCID mice xenografted with the human glioblastoma U87MG cell line (see Figure 5—figure supplement 1a). We also confirmed that IL-15 infusion increases the number of CD3$^-$/NK1.1$^+$ cells (NK cells) in the brains of tumor-implanted mice (Figure 5—figure supplement 1b). These cells were mostly activated, being CD69$^+$ and expressing granzyme B and IFN-γ (Figure 5—figure supplement 1c–e). In accordance, chronic infusion of IL-15 increased IFN-γ levels in the brains of glioma-bearing mice. The IL-15-induced increase of IFN-γ depends on NK cells, being absent in mice treated with an anti-NK 1.1 Ab (Figure 5a). We then investigated the effect of IL-15 on myeloid cell phenotype. CD11b$^+$ cells isolated from the brains of glioma-bearing mice treated with IL-15, as described in the scheme of Figure 5a, showed a phenotype shifted towards the pro-inflammatory state, with significantly increased expression of pro- and reduced expression of anti-inflammatory genes (Figure 5b). Similar results were obtained when animals were injected with CT-2a cells (Figure 5—figure supplement 1f). In accordance with this modulation of gene expression, IL-15 treatment also reduced the amount of CD68$^+$ and F4/80$^+$/CD68$^+$ cells in the tumor mass (Figure 5c), indicative of reduced activation of GAMs. Interestingly, the IL-15-dependent modulation of gene expression was completely absent in glioma-bearing mice in which NK cells were depleted (Figure 5d), further supporting the hypothesis that IL-15 modulates CD11b$^+$ cell phenotype by acting through NK cells.

The role of IL-15 in modulating the effect of EE was also confirmed with glioma Il15ra$^{-/-}$ mice: in these animals, we observed increased tumor size in SE, and a smaller effect of reducing tumor size in EE (Figure 5e). In addition, in CD11b$^+$ cells isolated from the brain of Il15ra$^{-/-}$ mice, the modulation of gene expression induced by EE was fully abolished (Figure 5f), thus demonstrating a key role for IL-15 signaling in modulating GAMs' phenotype.

## Brain-derived growth factor produced in the brain of EE mice modulates NK cell activation through IL-15

BDNF expression markedly increases during exposure to an EE (Sale et al., 2014). We investigated the possible effects of BDNF on glioma biology, demonstrating that the infusion of BDNF in mouse brain induces IL-15 production by CD11b$^+$ cells (Figure 5g). In line with the hypothesis that BDNF is a key neurotrophin in mediating the effects of EE, we observed that Bdnf$^{-/+}$ EE mice have no increased production of IL-15 in CD11b$^+$ cells (Figure 5h). We also analyzed BDNF-induced IL-15

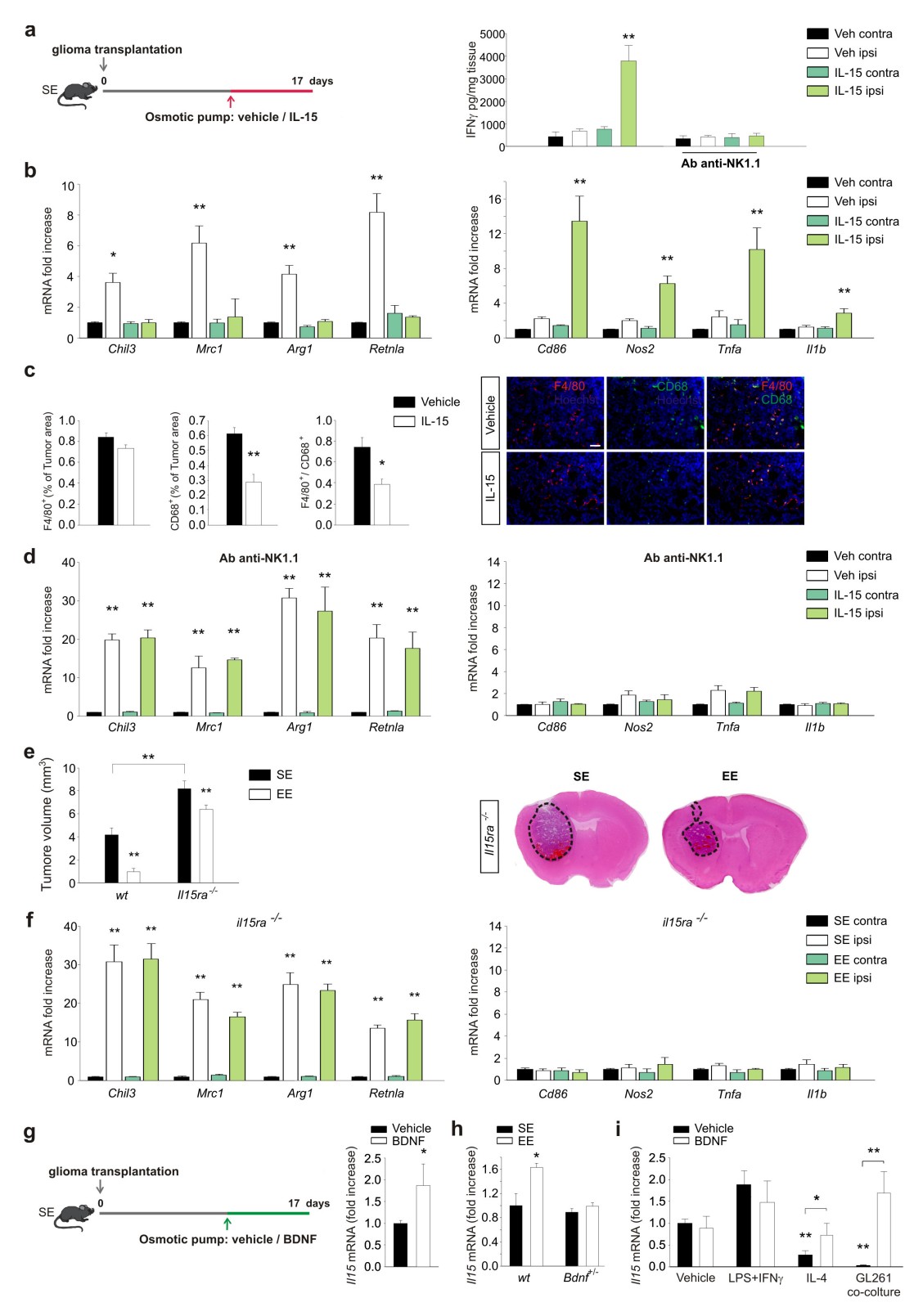

**Figure 5.** IL-15 is involved in the effects of EE on brain tumor. (a) Left: mice housed in SE were infused for 7 days in the striatum, with vehicle or IL-15 through micro-osmotic pumps, starting 10 days after GL261 cell transplantation, as described in the scheme. Right: expression of IFN-γ in CLH and ILH of GL261-bearing mice treated with vehicle or Ab anti-NK1.1 as indicated, implanted with osmotic pumps releasing IL-15 or vehicle (n = 5, **p<0.01, one-way ANOVA). (b) RT-PCR of anti- (*Chil3*, *Mrc1*, *Arg1*, *Retnla*) and pro-inflammatory (*Cd86*, *Nos2*, *Tnfa*, *Il1b*) genes in CD11b⁺ cells sorted from ILH

*Figure 5 continued on next page*

*Figure 5 continued*

and CLH of GL261-bearing mice treated with IL-15 or vehicle. Data are expressed as mean ± S.E.M., *p<0.05 **p<0.01 versus CLH, one-way ANOVA; n = 4. (c) Myeloid cell activation (CD68) and infiltration (F4/80) in glioma mass in mice treated with IL-15 or vehicle, as shown in (a), analyzed at the end of treatment (17 days after glioma transplantation). Graph bars represent the mean (± S.E.M.) area expressed as percentage of total tumor area. Representative immunofluorescences are shown on the right (scale bar, 100 µm) (**p=0.002 *p=0.011 Student's *t*-test; n = 4 mice per conditions). (d) RT-PCR of anti- and pro-inflammatory genes in CD11b$^+$ cells isolated from ILH and CLH of GL261-bearing mice in vehicle and Ab-anti-NK1.1-treated mice, upon IL-15 or vehicle infusion, as shown in (a). Data are expressed as the mean ± S.E.M., **p<0.01 versus CLH, one-way ANOVA; n = 4. (e) Analysis of tumor volumes (expressed as mm$^3$ ± S.E.M.) in *wt* or *Il15ra$^{-/-}$* mice, housed in SE or EE; n = 5; **p<0.01, two-way ANOVA). Representative coronal sections are shown on the right. (f) RT-PCR of anti- and pro-inflammatory genes in CD11b$^+$ cells sorted from ILH and CLH in *Il15ra$^{-/-}$* mice, housed in SE or EE. Data are the means ± S.E.M., **p<0.01 versus CLH, one-way ANOVA; n = 4–5. (g) RT-PCR of *Il15* mRNA in CD11b$^+$ cells sorted from ILH and CLH of GL261-bearing mice treated with BDNF or vehicle. Data are the mean ± S.E.M., *p<0.05 versus CLH, one-way ANOVA; n = 5. Above: scheme of striatal infusion of vehicle or BDNF with micro-osmotic pumps starting 10 days after glioma cell transplantation and lasting 7 days, in SE mice. (h) RT-PCR of *Il15* mRNA in CD11b$^+$ cells isolated from ILH and CLH in *wt* or *Bdnf$^{+/-}$* GL261 bearing mice, housed in SE or EE. Data are the mean ± S.E.M., *p<0.05 versus *CLH*, one-way ANOVA, n = 4. (i) RT-PCR of *Il15* mRNA in primary mouse microglia stimulated with vehicle, LPS + IFNγ or IL-4 or in co-culture with GL261 for 24 h, in the presence or absence of BDNF. Data are expressed as the mean ± S.E.M., *p<0.05 **p<0.01, one-way ANOVA, n = 10.

DOI: https://doi.org/10.7554/eLife.33415.011

The following figure supplements are available for figure 5:

**Figure supplement 1.** Effect of IL-15 treatment on tumor size, gene expression of myeloid cells and NK cell activation in EE mice injected with different glioma cells.

DOI: https://doi.org/10.7554/eLife.33415.012

**Figure supplement 2.** Gene expression in glioma cells and myeloid cells treated with BDNF.

DOI: https://doi.org/10.7554/eLife.33415.013

expression in vitro, using primary microglial cell cultures obtained after stimulation with vehicle, with LPS + IFN-γ or with IL-4, or co-cultured with GBM. We observed that BDNF induced IL-15 production by microglia only upon treatment with IL-4 or co-culture with glioma cells (*Figure 5i*). Control experiments confirmed that glioma cells were not the source of IL-15 (*Figure 5—figure supplement 2a*). In vitro experiments, performed to evaluate whether BDNF could directly modulate the pro- and anti-inflammatory gene expression by microglia, demonstrate no direct effects (*Figure 5—figure supplement 2b*).

## Role of peripheral monocytes in EE: depletion experiments

To evaluate the differential contribution of brain resident- versus infiltrating-myeloid cells in modulating the effects of EE on CD11b$^+$ cell phenotype and tumor size, we first analyzed CD11b$^+$/Ly6c$^+$-monocyte infiltration in the whole brain. We report a reduction in the frequency of CD11b$^+$/Ly6c$^+$ cells expressed as a percentage of infiltrating CD45$^+$ cells in the brain of EE mice (*Figure 6a*). Mice were treated with clodronate-containing liposomes to deplete peripheral phagocytes (*Van Rooijen and Sanders, 1994*). The efficacy of depletion was verified by FACS analysis. We observed a significant reduction in the extent to which the CD11b$^+$/Ly6c$^+$ cell population infiltrates the brain, and in the number of F4/80$^+$/Ly6c$^+$cells in the spleen (*Figure 6—figure supplement 1*). In these mice, the housing conditions induced a minor variation of gene expression: in particular, in SE, the CD11b$^+$ cells extracted from the ILH expressed all the anti-inflammatory markers analyzed except for CD206 (consistent with a selective peripheral phagocyte depletion). In EE mice, we observed a general decrease of anti-inflammatory genes accompanied by a significant increase of pro-inflammatory genes (*Figure 6b*). We also evaluated the effect of peripheral phagocyte depletion on tumor size (*Figure 6c*). In these mice, even in SE, tumor size is reduced, as expected considering the role of infiltrating cells in favoring tumor growth. In EE, clodronate-treated animals had a smaller tumor with respect to those in SE (*Figure 6c*), although the reduction was less than that in control mice treated with empty liposomes. (reduction of tumor volume in EE: 75.6 ± 6.6% for empty liposomes; 45.8 ± 9.6% for clodronate-filled liposomes, *p<0.05), suggesting the involvement of peripheral phagocytes in EE-induced tumor size reduction.

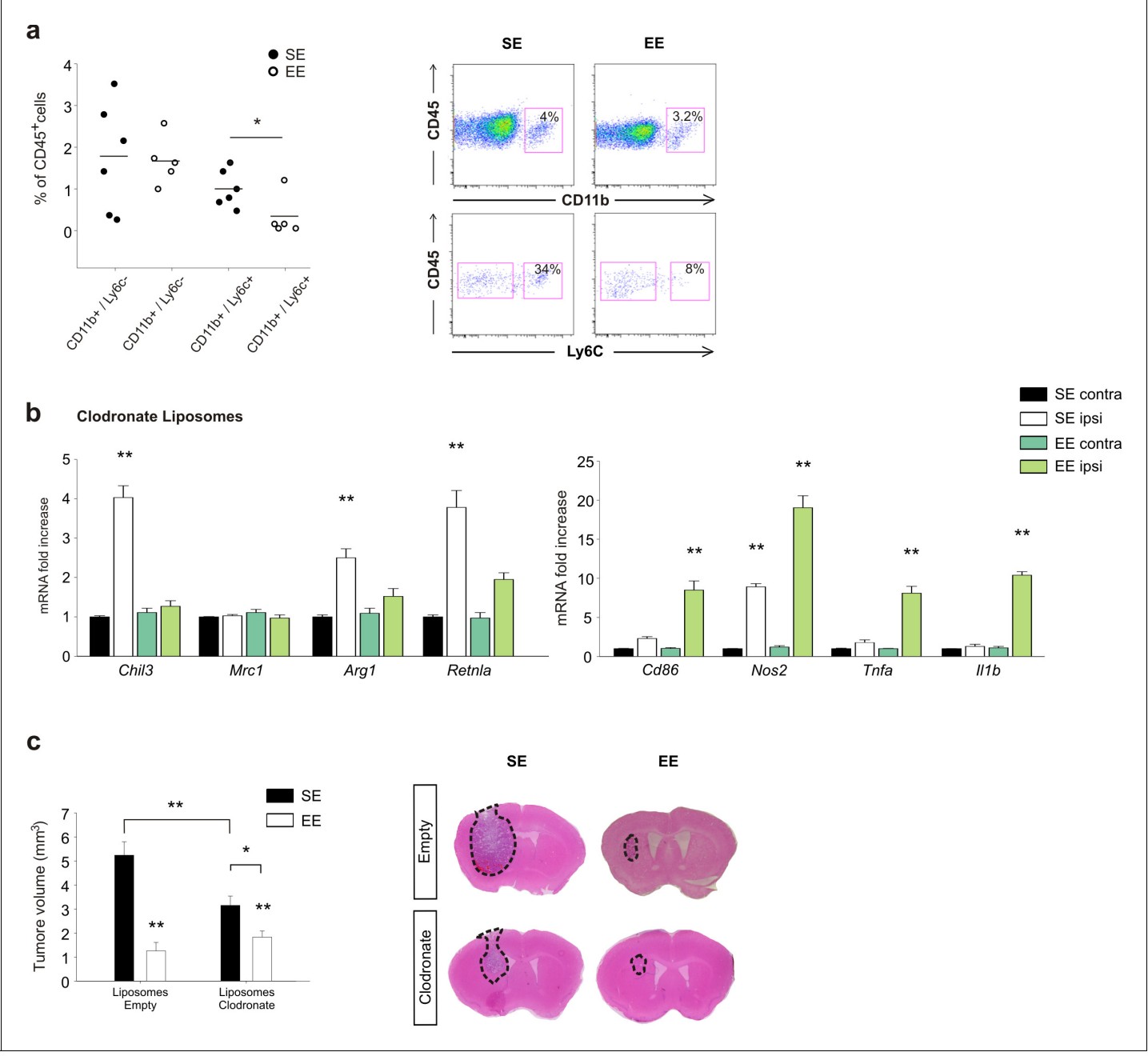

**Figure 6.** Effects of EE on mice treated with clodronate liposomes. (**a**) Percentage of CD11b[+] and Ly6c[+] cells in the total CD45[+] cell population obtained from the brain of EE or SE GL261-bearing mice (n = 5–6; *p=0.012, Student's *t*-test). Representative FACS analyses are shown below. (**b**) RT-PCR of anti- and pro-inflammatory genes in CD11b[+] cells sorted from ILH and CLH in empty or clodronate-filled liposome-treated mice, housed in SE or EE. Data are the mean ± S.E.M., **p<0.01 versus CLH by one-way ANOVA; n = 4. (**c**) Analysis of tumor volume (expressed as mm[3] ± S.E.M.) in liposome filled with clodronate or empty liposomes-treated GL261-bearing mice, housed in SE or EE; n = 5; *p<0.05 **p<0.01, two-way ANOVA). Representative coronal section are shown on the right.

DOI: https://doi.org/10.7554/eLife.33415.014

The following figure supplement is available for figure 6:

**Figure supplement 1.** Representative FACS analysis to verify CD45+/Ly6c + cell depletion in the brains of GL261-bearing mice housed in EE or SE, upon clodronate liposome treatment; and F4/80+/Ly6c + cells in the spleen.

DOI: https://doi.org/10.7554/eLife.33415.015

## Discussion

Myeloid cells have key roles in glioma progression. In different mouse models of glioma, GAMs acquire a predominant anti-inflammatory phenotype, and 'reeducation' of these cells towards a pro-inflammatory, anti-tumor phenotype successfully reduces tumor size and increases the mean survival times of tumor-bearing animals (*Mieczkowski et al., 2015*). GAMs, however, cannot be simply considered as M2-like cells, as they assume a plethora of different phenotypes, partially overlapping with the classical M1- or M2-macrophages, but in large part unique to GAMs (*Szulzewsky et al., 2015*). Conversely, in human GAMs, genes associated with immune pathways are not upregulated (*Szulzewsky et al., 2016*), suggesting that additional models of GBM must be considered before translating the results obtained in mouse models to patients. Nevertheless, the importance of GAMs in brain tumors has been recognized in humans, highlighting the need for further investigations, and also the need to explore different approaches. It is known that living in environments that are enriched with sensorial, physical and social stimuli can modulate the brain microenvironment, affecting the levels of hormones involved in feeding behavior or linked to the hypothalamic-pituitary axis, such as adiponectin, norepinephrine, BDNF and glucocorticoids (*Cao et al., 2010*). The external environment affects microglial cell proliferation in a brain-region-specific manner (*Ehninger and Kempermann, 2003*; *Ehninger et al., 2011*). Living environment also modifies brain-infiltrating myeloid cells in a mouse model of depression, in which the expression of pro-inflammatory genes is high (*Chabry et al., 2015*), affecting the response to antidepressant treatment (*Alboni et al., 2016*). Finally, it changes microglial number and morphology in a mouse model of Alzheimer's disease (*Rodríguez et al., 2015*). However, the mechanistic links between the macroenvironment modifications and myeloid cell regulation are lacking. Among the potential candidates for this communication is BDNF, which is produced in the brain upon environmental stimuli, as well as other molecules that are secreted by peripheral and brain-infiltrating immune cells.

To evaluate these hypotheses, we first investigated the effects of EE on GAM phenotype and the possible role played by NK cells. Contrasting studies described either positive or no effects of EE on tumor progression (*Cao et al., 2010*; *Nachat-Kappes et al., 2012*; *Garofalo et al., 2015*; *Westwood et al., 2013*). The reason for these differences is not clear, and additional experimental approaches such as analyses of the animal microbiota could be useful. It is indeed possible that central and peripheral signals activated in EE-housed mice could modulate the microbial composition of the gut, in a bidirectional communication with the different tumor microenvironments (*Goldszmid et al., 2015*). Clinical studies demonstrate that specific distressing stimuli, such as depression, feelings of loneliness and lack of social relationships, represent important risk factors for cancer development and progression (*Armaiz-Pena et al., 2009*). Herein, we demonstrate that housing animals in EE deeply modifies GAM phenotype, in particular microglial phenotype, as shown by Affymetrix microarray gene expression profiling, GO analyses and RT-PCR analyses. Microglia, and in general CD11b+ myeloid cells, isolated from the brain of glioma-bearing mice housed in EE, show increased expression of pro- and reduction of anti-inflammatory genes, indicative of an anti-tumor phenotype (*Hambardzumyan et al., 2016*). EE also modifies CD11b+ cell morphology, increasing the length and the number of cell branches, the speed of process movement towards ATP (which mimics an injury signal) and the expression of *P2ry12* mRNA, thus suggesting the re-establishment of a more efficient patrolling activity of these cells (*Haynes et al., 2006*). P2RY12 is specifically expressed by microglia (*Butovsky et al., 2014*) and associated with ATP-dependent process patrolling (*Ohsawa et al., 2010*) and better survival of patients with astrocytoma (*Zhu et al., 2017*). In EE, IBA1+ cells infiltrating the tumor mass also have reduced cell body roundness as well as increased phagocytic vacuoles, indicating the acquisition of a more ramified morphology with higher phagocytic activity (*Sierra et al., 2013*).

Having previously shown that NK cells are involved in mediating the effect of EE on glioma progression (*Garofalo et al., 2015*), we now investigated the role of NK cells in EE-induced phenotypic switch of myeloid cells. Other studies also have correlated microglia/macrophage activation state with NK cell activity (*Kmiecik et al., 2014*). We demonstrate that NK cell depletion completely abolishes the effect of EE on pro- and anti-inflammatory gene expression in CD11b+ cells. Moreover, upon housing in EE, there was a significant increase of direct contacts between glioma and NK cells, although the relative distance between GFP+ myeloid cells and NK cells did not change, in agreement with the observations of an activated phenotype of NK cells in EE. The key role of NK cells as

mediators of the effects of EE on myeloid cells could be considered to be a little surprising, given the low frequency of NK cells in brain tumor (*Kmiecik et al., 2013*). However, we confirmed that, in EE, NK cells more efficiently colonize the brain, increasing their distance from CD31$^+$ endothelial cells, degranulate against glioma cells, and produce increased amounts of IFN-γ. Experiments in which IFN-γ was neutralized in glioma-bearing mice confirmed the crucial role of this cytokine as an effector of NK cell activity. IFN-γ has antitumor activities, increases tumor immunogenicity, and inhibits angiogenesis and proliferation in brain tumors too (*Duluc et al., 2009*). We demonstrated that depletion of IFN-γ in glioma-bearing mice abolished the effect of EE on the phenotypic switch of myeloid cells, reduced the environmental effect on tumor volume and abolished the concomitant increase of mice survival. Depletion of IFN-γ also affected the reduction of angiogenesis and tumor cell proliferation induced by EE. Analyzing the effect of IFN-γ in human GBM tissue, we reported the modulation of inflammatory gene expression in myeloid cells, thus demonstrating that IFN-γ also influences human myeloid cell phenotype.

It is known that pro-inflammatory cytokines produced by myeloid cells, such as IL-12 and IL-15, stimulate the production of IFN-γ by NK cells (*Strengell et al., 2002*). We demonstrated that, in glioma-bearing mice, brain infusion of IL-15: i) increased the frequency of infiltrating NK cells in the ILH, ii) increased the frequency of NK cells producing IFN-γ, and iii) mimicked the effect of EE on the switch to an inflammatory gene expression in myeloid cells. In mice depleted of NK cells, IL-15 did not induce this gene expression switch, thus demonstrating the importance of NK cells in mediating the effect of IL-15 on myeloid cells. In addition, IL-15 administration reduced the number of infiltrating F4/80$^+$/CD68$^+$ cells in the tumor mass, in line with the hypothesis of a preferential NK-myeloid cell communication mediated by IL-15 (*Waldmann, 2006*). In EE mice lacking IL-15Rα, we failed to observe the modulation of inflammatory genes, and tumor size did not decrease. These data demonstrate the crucial role played by IL-15 in mediating the cascade of pathways activated by NK cells and IFN-γ in glioma-bearing mice housed in EE. Moreover, they support the hypothesis of a therapeutic usage of IL-15, administered alone or in combination with IL-15Ra, in brain tumors (*Mathios et al., 2016*). We also analyzed the differential expression of inflammatory genes in microglia (CD11b$^+$/CD45$^{low}$/Ly6c$^-$/Ly6g$^-$ cells) and infiltrating monocytes (CD11b$^+$/CD45$^+$/Ly6c$^+$/Ly6g$^-$ cells) isolated from the ILH and CLH of glioma-bearing mice. We demonstrated the absence of CD11b$^+$/CD45$^+$/Ly6c$^+$/Ly6g$^-$cells in the CLH, indicating a specific localization of infiltrating monocytes in the tumor region. We also described that both microglia and infiltrating myeloid cells express, in the tumor region, higher levels of anti-inflammatory genes, whereas their expression was reduced in both cell populations in EE. In contrast, the increased expression of pro-inflammatory genes upon housing in EE is almost restricted to microglia, demonstrating a major role for these cells in antitumor activity. Peripheral phagocyte depletion with clodronate-filled liposomes confirms the major role played by microglia in reducing tumor size in EE, whereas peripheral phagocytes control tumor size in SE. Altogether, these results indicate that brain-resident phagocytes are themselves targets of environmental stimuli, together with monocytes, although there are some differences in EE-induced phenotype modification.

To establish a mechanistic link between the variations in the macroenvironment and the modulation of myeloid cell phenotype, we considered BDNF as a potential mediator, because its brain levels increase upon psychosocial and physical enrichment (*Branchi et al., 2004*). We demonstrate that BDNF stimulated the production of IL-15 in the brain of glioma-bearing mice, and specifically in microglia upon challenge with IL-4 or co-culture with glioma cells. In summary, our results suggest that the neurotrophins produced in the brain of glioma-bearing EE mice activate a virtuous cycle starting with the production of IL-15 by microglial cells that, in turn, stimulates NK cells to produce IFN-γ, with effects on myeloid cell phenotype, switching them towards an anti-tumor state (see *Figure 7*), which explains the protective effects of the environment.

## Materials and methods

### Materials

Salts, glucose, adenosine 5'-triphosphate magnesium salt (Mg-ATP), hematoxylin, eosin, Percoll, BSA and deoxyribonuclease I were from Sigma-Aldrich (Milan, Italy). Transwell inserts were from BD Labware (Franklin Lakes, NJ). F4/80 (#NB300-141, RRID:AB_2246477) and NKp46 (M100) (#sc-

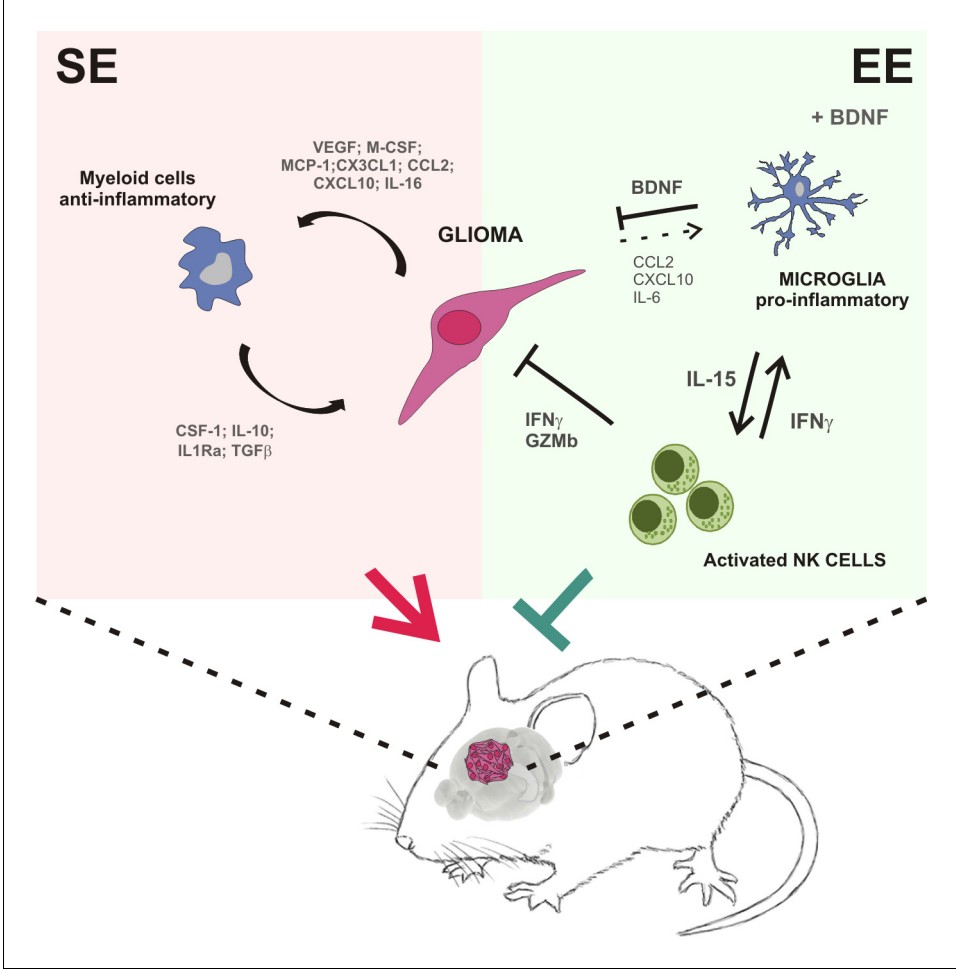

**Figure 7.** Summary of the events activated in EE-housed mice with glioma.
DOI: https://doi.org/10.7554/eLife.33415.016

---

18161, RRID:AB_2149152) antibodies (Abs) were from Santa Cruz Biotechnology (Santa Cruz, CA). LPS is from Immunotools (Friesoythe, Germany). CD31 (#3568S, RRID:AB_10694616) and Ki67 (#12202, RRID:AB_2620142) Abs were from Cell Signaling (Danvers, MA). CD68 Ab (#MCA1957T, RRID:AB_322219) was from AbD Serotec (Oxford, UK). Secondary Abs were from DAKO (Milan, Italy). Culture media, fetal bovine serum (FBS), goat serum, penicillin G, streptomycin, glutamine, Na pyruvate, Thermo Script RT–PCR System and Hoechst (#33342, RRID:AB_10626776) were from GIBCO Invitrogen (Carlsbad, CA, USA). BDNF, IL-15 and IFN-γ were from Immunological Sciences (Rome, Italy). IFN-γ ELISA kit was from eBioscience (San Diego, CA). Microbeads CD11b+ were from Miltenyi Biotec (Bologna, Italy). RNeasy Mini Kit was from Qiagen (Hilden, Germany). CD45, CD69, CD133, NK1.1 Abs, IL-15 were from eBioscience Inc., (San Diego, CA). Rat anti-mouse CD16/CD32 Ab was from BD Pharmingen (Milan Italy), and Rabbit anti-Iba1 from Wako (VA, USA). Osmotic pump (Alzet model 1007D, 100 ml; 0.5 ml/h), cannula (Azlet brain infusion kit 3) were from Charles River, Italy. Rat anti–IFN-γ monoclonal antibody (clone: XMG1.2, BioXcell), clodronate and empty liposomes were from Encapsula NanoSciences (Brentwood, TN).

## Mice and cell lines

Experiments described in the present work were approved by the Italian Ministry of Health in accordance with the guidelines on the ethical use of animals from the European Community Council Directive of September 22, 2010 (2010/63/EU). We used C57BL/6 (WT) and CB17/Icr-Prkdcscid/IcrIcoCrl (SCID) (RRID:IMSR_RBRC02771) mice from Charles River Laboratories; B6.129S4-Bdnf^tm1Jae/J

(*Bdnf*$^{+/-}$) (RRID:IMSR_JAX:002266), B6.129×1-Il15ra$^{tm.1Ama}$/J (*Il15ra*$^{-/-}$) (#3702076, RRID:MGI: 3702076) and heterozygous *Cx3cr1*$^{+/GFP}$ from The Jackson Laboratory. We always used two-month-old male mice. The GL261 glioma cell line (RRID:CVCL_Y003) (kindly provided by Dr. Serena Pellegatta, Istituto di Ricovero e Cura a Carattere Scientifico, Besta, Milan, Italy) and GL261-CD133$^+$ were cultured in DMEM supplemented with 20% heat inactivated FBS, 100 IU/ml penicillin G, 100 μg/ml streptomycin, 2.5 μg/ml amphotericin B, 2 mM glutamine, and 1 mM sodium pyruvate. The U87MG (RRID:CVCL_0022) and CT-2a cell lines (established by Thomas N. Seyfried) and the primary murine microglia were cultured in DMEM supplemented with 10% FBS.

None of the cell lines used was in the database of commonly misidentified cell lines maintained by ICLAC and NCBI Biosample. The cell lines were authenticated by STR profiling by the vendors and tested in our laboratory for mycoplasma contamination (negative).

## Environmental Enrichment (EE)

Three-week-old male mice (C57BL/6, *Cx3cr1*$^{+/GFP}$, *Il15ra*$^{-/-}$ or *Bdnf*$^{+/}$) were either housed in standard environment (SE) or EE. Mice exposed to SE were housed in pairs, in standard cages (30 cm × 16 cm × 11 cm), whereas mice exposed to EE were housed 10 in a cage (36 cm × 54 cm × 19 cm), in the presence of an assortment of objects, including climbing ladders, seesaws, running wheel, balls, plastic, and wood objects suspended from the ceiling, paper, cardboard boxes, and nesting material. Toys were changed every 2 days, and the bedding was changed every week. Both EE and SE groups received an identical type of rodent chow and water ad libitum and were kept on a 12 h light/dark cycle.

The detailed housing conditions are described in *Table 1*. Mice were kept in EE or SE for 5 weeks before brain implantation of human or murine glioma cells, then were left in their original cages for an additional 17 days. To cross validate our results, we performed preliminary experiments to evaluate the effects of various EE factors on tumor (GL261) volume: (i) housing mice in a different room; (ii) changing the person who manipulated the animals; (iii) handling mice in SE the same number of times as those in EE; and (iv) housing mice in EE before tumor injection and then in SE.

## Isolation of CD11b-positive cells and extraction of total RNA

Glioma bearing C57BL/6, *Il15ra*$^{-/-}$ or *Bdnf*$^{+/-}$ mice were anesthetized and decapitated. Brains were removed, brain tissues were cut into small pieces and single-cell suspension was achieved in Hank's balanced salt solution (HBSS). The tissue was further mechanically dissociated using a glass wide-tipped pipette and the suspension was applied to a 30 μm cell strainer (Miltenyi Biotec). Cells were processed immediately for MACS MicroBead separation. CD11b$^+$ cells were magnetically labelled with CD11b MicroBeads. The cell suspension was loaded onto a MACS Column placed in the magnetic field of a MACS Separator and the negative fraction was collected. After removing the magnetic field, CD11b$^+$ cells were eluted as a positive fraction. Live CD11b$^+$ cells were assessed by immunofluorescence and flow cytometry (FACS). After sorting the positive and negative fractions, total RNA was isolated with RNeasy Mini Kit, and processed for real-time PCR. The quality and yield of RNAs were verified using the NANODROP One system (Thermo Scientific).

## Isolation of CD11b$^+$ cells from human GBM

Tumor specimens obtained from adult patients with GBM who gave informed consent to the research proposals (IRCCS Neuromed) were dissociated and treated for 24h with IFN-γ (20 ng/ml) or vehicle. Tissues were then processed as described above to isolate CD11b$^+$ cells and mRNAs were analyzed by RT-PCR for gene expression.

## Preparation of CD133$^+$GL261 cells

Single-cell suspensions from cultured GL261 cells were incubated for 10 min at 4°C with rat anti-mouse CD16/CD32 Ab (1:250) and for 30 min at 4°C with CD133 Ab (1:75). The CD133$^+$ GL261 cell population was isolated using phycoerythrin anti-mouse CD133 Ab using a BD FACS AriaII (BD Biosciences). The purity of cell population was verified using flow cytometry (~90%). Sorted CD133$^+$ GL261 cells were maintained in DMEM with 20 ng ml$^{-1}$ fibroblast growth factor-2, 20 ng ml$^{-1}$ EGF and heparin 10 U ml$^{-1}$, for 24 before brain transplantation.

**Table 1.** Housing conditions in enriched environment (EE).

| Variables | Value |
|---|---|
| EE cage size (cm) | 36 x54 x19 |
| EE cage composition | polycarbonate |
| Control cage | polycarbonate |
| EE floor space/mouse (cm 2) | 195 |
| # Mice/EE cage | 10 |
| Stimulating toys/objects in EE cage | 2 running wheels, tunnels, 2 refuges 1 swing, with nesting material |
| Objects varied regularly? | Yes |
| Strain of mice | C57BL/6 |
| Sex of mice | Male |
| EE, control cages in same room? | Yes |
| Lighting | 12 hours on/off |
| Temp (degrees C) | 22 ± 1 |
| Bedding | Sawdust |
| Humidity control? | No |
| Cleaning schedule | Once a week |
| Food based on wheat, oats, meat, soy and milk? | Yes (14% protein, 5% fat, 3041 kcal ME/kg) |
| Microbiota endemic in animal facility | Norovirus, Helicobacter |
| Age of mice put in cage initially | 3 weeks |
| # weeks habituation | 5 weeks |
| Tumor injected | GL261 |
| Route injected | intrastriatal |
| # Cells injected | $7.5\times10^4$ |
| Mouse handling frequency | Every 3 days |
| Statistical significance in tumor size? | Yes |

DOI: https://doi.org/10.7554/eLife.33415.017

## Real-time PCR

CLH or ILH of injected mice, housed in EE or SE, were lysed in Trizol reagent for isolation of RNA. Reverse transcription reaction of brain hemispheres or CD11b$^+$ and CD11b$^-$ fractions collected by MACS was performed in a thermocycler (MJ Mini Personal Thermal Cycler; Biorad) using IScript TM Reverse Transcription Supermix (Biorad) according to the manufacturer's protocol, under the following conditions: incubation at 25°C for 5 min, reverse transcription at 42°C for 30 min, inactivation at 85°C for 5 min. Real-time PCR (RT-PCR) was carried out in a I-Cycler IQ Multicolor RT-PCR Detection System (Biorad) using SsoFast EvaGreen Supermix (Biorad) according to the manufacturer's instructions. The PCR protocol consisted of 40 cycles of denaturation at 95°C for 30 s and annealing/extension at 60°C for 30 s. For quantification analysis, the comparative Threshold Cycle (Ct) method was used. The Ct values from each gene were normalized to the Ct value of GAPDH in the same RNA samples. Relative quantification was performed using the $2^{-\Delta\Delta Ct}$ method (*Schmittgen and Livak, 2008*) and expressed as fold change in arbitrary values. The following primers were used.

| Gene | Species | Primer forward (5'—3') | Primer Reverse (3'—5') |
|---|---|---|---|
| *Arg1* | Mouse | CTCCAAGCCAAAGTCCTTAGAG | AGGAGCTGTCATTAGGGACATC |
| *Cd86* | Mouse | AGAACTTACGGAAGCACCCA | GGCAGATATGCAGTCCCATT |
| *Mrc1* | Mouse | CAAGGAAGGTTGGCATTTGT | CCTTTCAGTCCTTTGCAAGT |
| *Retnla* | Mouse | CCAATCCAGCTAACTATCCCTCC | ACCCAGTAGCAGTCATCCCA |

*Continued on next page*

*Continued*

| Gene | Species | Primer forward (5'—3') | Primer Reverse (3'—5') |
|------|---------|------------------------|------------------------|
| Gapdh | Mouse | TCGTCCCGTAGACAAAATGG | TTGAGGTCAATGAAGGGGTC |
| Il1b | Mouse | GCAACTGTTCCTGAACTCAACT | ATCTTTTGGGGTCCGTCAACT |
| Nos2 | Mouse | ACATCGACCCGTCCACAGTAT | CAGAGGGGTAGGCTTGTCTC |
| Tnfa | Mouse | GTGGAACTGGCAGAAGAG | CCATAGAACTGATGAGAGG |
| Chil3 | Mouse | CAGGTCTGGCAATTCTTCTGAA | GTCTTGCTCATGTGTGTAAGTGA |
| Il15 | Mouse | CATCCATCTCGTGCTACTTGTGTT | CATCTATCCAGTTGGCCTCTGTTT |
| P2ry12 | Mouse | CCTGTCGTCAGAGACTACAAG | GGATTTACTGCGGATCTGAAAG |
| Cd163 | Human | TCTGGCTTGACAGCGTTTG | TGTGTTTGTTGCCTGGATT |
| Mmp12 | Human | AGGAATCGGGCCTAAAATTG | TGCTTTTCAGTGTTTTGGTGA |
| Tgfb | Human | CCAACTATTGCTTCAGCTCCAC | GTGTCCAGGCTCCAAATGTAGG |
| Cxcl10 | Human | GTGGCATTCAAGGAGTACCTC | TGATGGCCTTCGATTCTGGATT |
| Nos2 | Human | CAGCGGGATGACTTTCCAA | AGGCAAGATTTGGACCTGCA |
| Il12a | Human | CTCCTGGACCACCTCAGTTTG | GGTGAAGGCATGGGAACATT |
| Gapdh | Human | CCCCTTCATTGACCTCAACTAC | GATGACAAGCTTCCCGTTCTC |

## Microarray gene expression profiling

Microglia were isolated from ILH of EE or SE mice 17 days after GL261 injection. Total RNA was extracted using the RNeasy kit (Qiagen, Germany). The amount and quality of the RNA were determined with RNA 6000 Nano Kit (Agilent Technologies, Santa Clara (CA)) and Bioanalyzer 2100 (Thermo Scientific, Germany). Microarray experiments were performed with 100 ng of total RNA and HT-MG-430 PM Affymetrix strip according to the manufacturer's User Guide for the GeneAtlas Personal Microarray System (Affymetrix, Santa Clara, CA, USA) . All microarray data analyses were performed in the R statistical environment and relevant Bioconductor software (RRID:SCR_001905). The raw data were pre-processed using RMA. Annotation of probe sets was performed with information provided in the Ensembl database. Because of the low number of samples, changes in gene expression levels were evaluated with Welsch *t*-test, and a high fold-change threshold (>4) was used to minimize the number of false positives. Genes were assigned to Gene Ontology terms according to the GO stats Bioconductor package, and Fisher exact test followed by Bonferroni correction were used to identify overrepresented GO terms.

## In vivo NK cell depletion

C57BL/6 mice were treated with Ab anti-NK1.1 (mAb PK136, 200 µg in 100 µl) by i.p. injections repeated with the following scheme: 2 days before glioma transplantation, and at day 0, 2 and 7 after glioma transplantation. At 7 and 17 days after glioma surgery, NK cell depletion from blood was verified as described (*Garofalo et al., 2015*).

## In vivo IFN-γ depletion

Mice housed in EE or SE were treated with a single dose of rat mAb anti–IFN-γ (XMG1.2, 2 mg) or control mAb isotype, by i.p. injection on the same day as glioma transplantation, and then every 5 days until the mice were sacrificed.

## Clodronate liposome administration

Depletion of peripheral phagocytes was performed using clodronate-filled liposomes or empty liposomes as control. C57BL/6 mice housed in EE or SE were injected with GL261 cells and treated with clodronate 2.5 mg/ml in 100 µl of liposomes. Liposomes were injected i.p. 4 days before, during and every 4 days after glioma transplantation until the mice were sacrificed. 7 and 17 days after glioma implantation, cell depletion was verified from brain and spleen sample sorting cells by FACS for CD11b/Ly6C or F4/80/Ly6C staining.

## Isolation of microglia or infiltrating monocytes

The CLH or ILH of glioma-bearing C57BL/6 mice were cut into small pieces and the tissue was mechanically dissociated using a wide-tipped glass pipette. The suspension was then passed through a 70 μm nylon cell strainer. CD11b$^+$ cells, isolated by MACS, were further selected for CD45$^{low}$, CD11b$^+$, Ly6c$^-$, Ly6g$^-$, or for CD45$^+$, CD11b$^+$, Ly6c$^+$, Ly6g– by FACS Aria II (BD Biosciences). Cell purity was verified by flow cytometry. After cell sorting, total RNA was isolated by RNeasy Mini Kit and processed for real-time PCR.

## NK cell degranulation assay

GL261 cells were seeded in a 96-well plate at $2 \times 10^3$ cells/well the day before the assay. Brain immune cells from SE and EE tumor-bearing mice were enriched by centrifugation on percoll 40%, washed in PBS and co-incubated for 4h at 37°C with GL261 or YAC-1 cell lines at 2: 1 effector:target ratio in complete RPMI 1640 medium (Hepes 10mM + Monensin 100 μM + Il-2 100U/well). FITC-conjugated anti-mouse CD107a or anti-IgG were added into appropriate wells 3h before the end of the assay. Cells were immunostained with fluorochrome-conjugated anti-CD3 Ab, anti-NK1.1 Ab, anti-CD45.2 Ab to identify NK cells and analyzed by FACS analysis.

## Measurement of IFN-γ by ELISA

After 17 days from glioma transplantation, ILH and CLH of EE or SE mice were disrupted with a homogenizer and were analyzed for IFN-γ content using a sandwich ELISA, following the manufacturer's instructions. Briefly, 96-well ELISA microplates were coated with anti-IFN-γ monoclonal Ab. Samples or IFN-γ standard were added at the appropriate dilution and incubated for 2h at room temperature. After careful washing, biotinylated goat anti-human IFN-γ was added to each well; horseradish-peroxidase was used as secondary Ab and optical density was read at 450 nm.

## Osmotic pump implantation

Ten days after glioma transplantation, 8-week-old male C57BL/6 or SCID mice were implanted with an osmotic pump (Alzet model 1007D, 100 μl; 0.5 μl/h; Charles River, Italy) in the right striatum. Before surgery, mice were anaesthetized and placed on the stereotaxic apparatus to implant a cannula (Alzet brain infusion kit 3) into the right striatum, through the same hole used for glioma injection. The cannula was sealed with dental cement and connected to the Alzet pump. Before surgery, the pumps and the tubing were incubated at 37°C overnight in a sterile saline solution for priming. The pump was then placed into a subcutaneous pocket in the dorsal region. The pumps were filled with vehicle (PBS), BDNF (60 ng/μl) or IL-15 (30 ng/μl), and infusion lasted 7 days. The BDNF and IL-15 doses were selected on the basis of previous in vitro experiments (*Garofalo et al., 2015*).

## Primary microglial cultures and co-cultures with glioma cells

Microglia cultures were obtained from mixed glia cultures derived from the cerebral cortices of post-natal day 0–1 (p0–p1) C57BL/6 mice, as described (*Lauro et al., 2010*). In brief, cortices were chopped and digested in 15 U/ml papain for 20 min at 37°C. Cells ($5 \times 10^5$ cells/cm$^2$) were plated on flasks coated with poly-L-lysine (100 mg/ml) in DMEM supplemented with 10% FBS, 100 U/ml penicillin, and 0.1 mg/ml streptomycin. After 7–9 days, cells were shaken for 2h at 37°C to detach and collect microglial cells. These procedures gave almost pure microglial cell populations.

Primary microglia cultures were co-cultured with the GL261 glioma cell line on 0.4 μm pore size polycarbonate membranes of transwells for 24 hr. Microglia were plated ($5 \times 10^4$ cells) at the bottom of the dish and GL261 cells ($8 \times 10^4$ cells) on the polycarbonate membrane.

## Microglial polarization

Primary microglia cultures were treated for 24h with LPS 100 ng/ml + IFN-γ 20 ng/ml, or IL-4 20 ng/ml to induce cell polarization, and then for an additional 24h in the presence or absence of BDNF 100 ng/ml.

## Isolation of leukocytes

Seventeen days after glioma transplantation, mice were perfused intra-cardially with PBS and the brain rapidly removed; hemispheres were separated, placed into 5 mL of ice-cold PBS containing

0.2% bovine serum albumin (BSA), 0.01 mol/L EDTA, and 1 mg/ml deoxyribonuclease I. The hemispheres were disrupted in a glass homogenizer and passed through a 70 μm nylon cell strainer (Becton Dickinson). The suspension was centrifuged at 400 g for 10 min at RT and the pellets re-suspended in 4 ml of 40% Percoll (Amersham Pharmacia Biotech). Percoll was prepared by dilution in Hanks' balanced salt solution (HBSS). The gradient was centrifuged at 1600 rpm for 20 min at RT; the cell pellet was subsequently collected (approximately 0.5 ml) and washed once with HBSS containing 10% FBS. After removal of contaminating erythrocytes, cells were counted, and 30–50 ml of whole blood cells were washed and re-suspended in staining buffer. Cells were then blocked by incubation in ice with anti-CD16/CD32 (24G2) to prevent nonspecific and Fc-mediated binding. After 10 min, a mixture of the appropriate antibodies was added. Cells were further incubated on ice for 30 min, washed and analyzed using a FACSCanto II (BDBiosciences). Data were elaborated using the FlowJo Version 7.6 software (Tree Star).

## Intracranial injection of glioma

Male C57BL/6, SCID, $Cx3cr1^{+/GFP}$, $Il15ra^{-/-}$ or $Bdnf^{+/-}$ mice were anesthetized with chloral hydrate (400 mg/kg, i.p.) and placed in a stereotaxic head frame. Animals were stereotactically injected with $7.5 \times 10^4$ GL261 or GL261-CD133$^+$ cells, $8 \times 10^4$ CT-2a cells, and $5 \times 10^4$ U87MG cells: a median incision of ~1 cm was made, a burr hole was drilled in the skull, and cells were injected 2 mm lateral (right) and 1 mm anterior to the bregma in the right striatum. Cell suspensions, in PBS (4 μl) were injected with a Hamilton syringe at a rate of 1 μl/min at 3 mm depth. After 17 days, animals were sacrificed for different analyses.

## Histopathological evaluation of tumor volume

After 17 days from glioma cells injection, brains were isolated for morphological evaluation of tissues and fixed in 4% buffered formaldehyde. Coronal brain sections (20 μm) were prepared by standard procedures and stained with hematoxylin and eosin. A section every 100 μm was collected, and the tumor area was evaluated using Image Tool 3.00.

## Survival analysis

Following injection of the glioma cells, mice were treated with Ab anti-IFN-γ or vehicle every 5 days and were monitored daily. The endpoint was defined by the lack of physical activity or death. The probability of survival was calculated using the Kaplan–Meier method, and statistical analysis was performed using a log-rank test.

## Immunostaining

Seventeen days after injection of glioma cells, mice were overdosed with chloral hydrate (400 mg/kg, i.p.) and then intra-cardially perfused with PBS; brains were then isolated and fixed in 4% formaldehyde and snap frozen. Cryostat sections (20 μm) were washed in PBS, blocked (3% goat serum in 0.3% Triton X-100) for 1 h at RT, and incubated overnight at 4°C with specific antibodies diluted in PBS containing 1% goat serum and 0.1% Triton X-100. The sections were incubated with the following primary Abs: anti-F4/80 (1:50), anti-CD68 (1:200), anti-Ki67 (1:200), anti-CD31 (1:200), and anti-Nkp46 (1:100). After several washes, sections were stained with the fluorophore-conjugated antibody and Hoechst for nuclei visualization and analyzed using a fluorescence microscope. For co-immunofluorescence, the secondary antibody was subsequently used. For F4/80 staining, coronal sections were first boiled for 20 min in citrate buffer (pH 6.0) at 95–100°C.

## Image acquisition and data analysis

Images were digitized using a CoolSNAP camera (Photometrics) coupled to an ECLIPSE Ti-S microscope (Nikon) and processed using MetaMorph 7.6.5.0 image analysis software (Molecular Device). Slices were scanned by consecutive fields of vision (x 10 objective lens) to build a single image per section. Data were expressed as area occupied by fluorescent cells versus total tumor area (by converting pixel to mm [*Graeber et al., 2002*]). For comparison between different treatments, at least 12 coronal sections per brain around the point of injection were analyzed.

## Slice preparation

Acute coronal brain slices were prepared from SE and EE *Cx3cr1$^{+/GFP}$* mice (*Jung et al., 2000*) injected with Tag-RFP GL261 cells (*Garofalo et al., 2015*), in chilled artificial cerebrospinal fluid (ACSF) containing (in mM): NaCl 125, KCl 2.3, CaCl$_2$ 2, MgCl$_2$ 1, NaHPO$_4$ 1, NaHCO$_3$ 26 and glucose 10. The ACSF was continuously oxygenated with 95% O$_2$, 5% CO$_2$ to maintain physiological pH. Coronal 250 μm slices were cut at 4°C with a vibratome (DSK, Kyoto, Japan), placed in a chamber containing oxygenated ACSF and allowed to recover for at least 1h at RT (24–25 °C). All recordings were performed at RT on slices submerged in ACSF and perfused (1 ml/min) with the same solution in the recording chamber under the microscope.

## Time-lapse imaging in brain slices

Time-lapse fluorescence determinations were acquired at RT using a customized digital imaging microscope. Excitation of GFP was achieved using a Cairn Research – OptoScan monochromator. Fluorescence was visualized using an upright microscope (Olympus) equipped with a 40x water-immersion objective and a digital 14 bit CCD camera system (Cool SNAP MYO, Photometrics). All the peripheral hardware controls, image acquisition and processing were achieved using Metafluor software (Molecular Device). A glass pipette containing ATP (2 mM) was placed in the center of the tumoral or peritumoral areas or in the *CL* striatum. Mg-ATP was pressure applied to the slices (100 ms; 5 p.s.i.) with a Pneumo pump (WPI). Changes in GFP fluorescence distribution were monitored by acquiring a fluorescent image every 10 s for 50 min. To quantify the speed of process rearrangement toward the pipette tip, we measured the increase of GFP fluorescence in a circular area centered on the pipette tip (10 μm radius). At each time point, the fluorescence increase in the area was calculated as $\Delta F = F\ F_0$, and then divided by $F_0$ ($\Delta F/F_0$, where $F_0$ is the average fluorescence before ATP puff), to normalize the difference in basal GFP fluorescence in slices from the two conditions. Slices were used from 2 to 7h after cutting.

## Whole-cell patch-clamp recordings

Visually identified GFP-expressing cells were patched in whole-cell configuration in the tumoral area, peritumoral area and CLH striatum. Micropipettes (4–5 MΩ) were usually filled with a solution containing (in mM): KCl 135, BAPTA 5, MgCl$_2$ 2, HEPES 10, and Mg-ATP 2 (pH 7.3 adjusted with KOH, osmolarity 290 mOsm). Voltage-clamp recordings were performed using an Axopatch 200B amplifier (Molecular Devices). Currents were filtered at 2 kHz, digitized (10 kHz) and acquired using Clampex 10 (Molecular Devices). The analysis was performed off-line using Clampfit 10 (Molecular Devices). The current/voltage (I/V) relationship of each cell was determined by applying voltage steps from −170 to +70 mV (ΔV = 10 mV) for 50 ms holding the cell at −70 mV between steps. Resting membrane potential and membrane capacitance were measured at the start of recording. Outward- and inward-rectifier K$^+$ current amplitude was evaluated after subtraction of the leak current by a linear fit of the I/V curve between −100 and −50 mV. Cells were considered as expressing the outward-rectifier K$^+$ current when the I/V relationship showed a rectification above −30 mV and the amplitude measured at 0 mV was at least 5 pA, after leak subtraction; similarly, cells showing a small inward rectification below −100 mV were classified as expressing the inward rectifier K$^+$ current when the subtracted current amplitude was at least 5 pA at −150 mV.

Membrane capacitance (Cm) of patched cells was estimated as the total charge (i.e., the current integral, Q step) mobilized in each cell by a 10 mV depolarizing step (V step): Q step/V step. Resting membrane potential (RP) and membrane resistance (Rm) were measured immediately after membrane rupture. Average values of Cm, Rm and RP are reported in *Table 2*.

## Process movement

Image stacks were processed using the ImageJ 'Manual Tracking' plug-in (http://imagej.nih.gov/ij/plugins/track/track.html) as previously described (*Pagani et al., 2015*). Briefly, stacks were initially background subtracted to optimize contrast. Track positions of single processes were transferred into a new coordinate system, in which the initial position of each track was set as the origin (x = 0, y = 0). For each moving process (i), with position vector Ri(t), the change in position from one frame to the next (ΔRi[t]) and the instantaneous velocity (vi[t]) were given by ΔRi(t)=Ri(t+Δt)−Ri(t), and vi(t) =ΔRi(t)/Δt respectively, where Δt is the elapsed time between the two frames. The mean velocity of

**Table 2.** Passive properties of patched microglia: membrane capacitance (Cm), resting membrane potential (RP) and membrane resistance (Rm) were measured as described in the methods.

| | Cm (pF) | Rm (MΩ) | RP (mV) | N |
|---|---|---|---|---|
| Tum SE | 26 ± 2 | 1.5 ± 0.1 | –45 ± 3 | 57 |
| Tum EE | 26 ± 1 | 2.1 ± 0.2 | –46 ± 2 | 64 |
| Peri SE | 27 ± 1 | 1.9 ± 0.2 | –43 ± 4 | 60 |
| Peri EE | 25 ± 1 | 2.2 ± 0.2 | –49 ± 3 | 57 |
| *CLH* SE | 18 ± 3 | 2.7 ± 0.4 | –53 ± 3 | 38 |
| *CLH* EE | 20 ± 1 | 2.4 ± 0.3 | –47 ± 4 | 27 |

DOI: https://doi.org/10.7554/eLife.33415.018

each process was calculated as <v>= dx/dt, expressed in µm/min, defining dx as the mean accumulated distance of each process i sampled within the time interval dt. Migration length of single processes was also analyzed using the 'Manual Tracking' plug-in tool. Quantitative distributions of track parameters were analyzed with Origin 7 (OriginLab Co.) software.

## Morphological analysis of myeloid cells and 3D reconstruction

Brain slices from perfused brains were analyzed by confocal microscopy and skeleton analysis to assess myeloid cell morphology using endogenous GFP signal. Twenty µm z-stacks were acquired at 0.5 µm intervals using an FV1000 laser scanning microscope (Olympus) at ×60 objective. Cell morphology was measured in tumoral and peritumoral region using a method adapted from that described by *Morrison and Filosa (2013)*. Maximum intensity projections for the GFP channel of each image were generated, binarized, and skeletonized using the Skeletonize 2D/3D plugin in ImageJ, after which the Analyze Skeleton plugin (http://imagej.net/AnalyzeSkeleton) was applied. The average branch number (process end points per cell) and length per cell were recorded for each image with a voxel size exclusion limit of 150 applied. The number of single and multiple junction points was additionally calculated to give an indication of branching complexity. The areas of the soma and scanning domain were measured for each cell. 3D reconstruction of NK cells, GL261 and myeloid cells was achieved by confocal microscopy analysis with a FV1200 (Olympus) Laser Scanning System, at 60x magnification. Myeloid and GL261 cells were detected using endogenous GFP and RFP signals respectively, NK cells were identified with an Alexa fluor 633 conjugated NK1.1 antibody. Acquisition files were then processed with ImageJ software for two-dimensional analysis. Three-dimensional reconstructions were generated with Imaris software (Bitplane, Zurich, Switzerland) and morphometric analysis of each reconstructed cell, both in acute slices and in perfused brain sections, was performed after surface and volume rendering. Distance plugin of Imaris software was applied to evaluate distances between surfaces of NK cells, GL261 and microglia.

## Electron microscopy

Immunoperoxidase staining for electron microscopy was performed as described previously (*Tremblay et al., 2010*; *Bisht et al., 2016*). Briefly, mice housed in EE or SE were anesthetized with sodium pentobarbital (80 mg/kg, i.p.) and perfused through the aortic arch with 0.1% glutaraldehyde in 4% paraformaldehyde. Only mice for which the perfusion was optimal were included in the study; brain sections containing the tumor were washed in PBS, quenched with 0.3% $H_2O_2$, then with 0.1% $NaBH_4$, washed in Tris-buffered saline (TBS; 50 mM at pH 7.4) containing 0.01% Triton X100, and processed freely floating for immunostaining. Sections were blocked in TBS containing 10% FBS, 3% BSA and 0.01% Triton X100, prior to overnight incubation at 4°C with primary antibody (rabbit anti-Iba1 1:1000). Sections were then incubated with secondary antibodies (biotin-conjugated goat anti-rabbit IgG 1:200) for 1.5h. Staining was amplified by the ABC Vecta stain system (Vector laboratories), and revealed using diaminobenzidine (DAB; 0.05%) and hydrogen peroxide (0.015%) in TBS for 5 min. Sections were further processed for electron microscopy as follows: they were post-fixed in 1% osmium tetroxide and dehydrated using increasing concentrations of ethanol and finally immersed in propylene oxide. Following dehydration, sections were impregnated with Durcopan resin (EMS) overnight at RT, mounted between ACLAR embedding films (EMS), and cured at

55°C for 72h. Specific regions of interest were excised and mounted on resin blocks for ultrathin sectioning.

Ultrathin (65–80 nm) sections were cut and collected on bare square mesh grids (EMS) using an ultramicrotome (Leica Ultracut UC7) and imaged at 80kV with a FEI Tecnai Spirit G2 transmission electron microscope. Profiles of neurons, synaptic elements, microglia, myeloid cells, astrocytes, oligodendrocytes, and myelinated axons were identified according to well-established criteria. Microglia were identified both by their Iba1 immunoreactivity and by their association with extracellular space, with distinctive long stretches of endoplasmic reticulum, and with small elongated nuclei (*Tremblay et al., 2010, 2012*). Myeloid cell bodies were photographed at various magnifications between 800x and 9300x using an ORCA-HR digital camera. In these images, we analyzed the size and shape of myeloid cells, ultrastructural features of microglia, contacts with NK and tumor cells, as well as phagocytic activity. All the analysis was performed by investigators blinded to the experimental conditions using FIJI.

## Statistical analysis

Data are shown as the mean ± SEM. Statistical significance was assessed by Student's *t*-test, one-way ANOVA or two-way ANOVA for parametrical data, as indicated; Holm–Sidak test was used as a *post hoc* test; Mann–Whitney Rank test and Kruskal–Wallis for non-parametrical data, followed by Dunn's or Tukey's *post hoc* tests. For multiple comparisons, multiplicity-adjusted *p*-values are indicated in the corresponding figures (*p 0.05, **p 0.01). For the Kaplan–Meier analysis of survival, the log-rank test was used. Statistical analyses comprising calculation of degrees of freedom were done using Sigma Plot 11.0, Imaris; Origin 7, and Prism 7 software. I/V plots, cumulative distribution plots, and fitted data points were constructed by linear or nonlinear regression analysis using Origin 7 software. Statistical significance for cumulative distributions was assessed with Kolmogorov-Smirnov test.

## Acknowledgements

We are grateful to Louis Samson for help with the electron microscopy experiments, to Julie-Christine Lévesque and Sachiko Sato at the Bio-Imaging Facility of CRCHU de Québec-Université Laval for technical assistance, and to Prof. Davide Ragozzino, Flavia Trettel and Francesca Grassi for discussions.

## Additional information

### Funding

| Funder | Grant reference number | Author |
|--------|------------------------|--------|
| Associazione Italiana per la Ricerca sul Cancro | AIRC2015 IG16699 | Cristina Limatola |
| Ministero dell'Istruzione, dell'Università e della Ricerca | PRIN 2015 | Cristina Limatola |
| CRCHU | Starting Grant | Eve Tremblay |
| European Commission | Euronanomed2: Nanoglio | Angela Santoni |
| Associazione Italiana per la Ricerca sul Cancro | AIRC2014 IG16014 | Angela Santoni |

The funders had no role in study design, data collection and interpretation, or the decision to submit the work for publication.

### Author contributions

Stefano Garofalo, Conceptualization, Data curation, Formal analysis, Investigation, Methodology, Writing—original draft; Alessandra Porzia, Data curation, Investigation, Methodology; Fabrizio Mainiero, Conceptualization, Supervision, Writing—review and editing; Silvia Di Angelantonio, Conceptualization, Data curation, Supervision, Investigation, Methodology, Writing—review and editing;

Barbara Cortese, Conceptualization, Data curation, Investigation; Bernadette Basilico, Karolina Stepniak, Data curation, Formal analysis, Methodology; Francesca Pagani, Data curation, Investigation; Giorgio Cignitti, Data curation, Formal analysis; Giuseppina Chece, Roberta Maggio, Julie Savage, Kanchan Bisht, Data curation, Methodology; Marie-Eve Tremblay, Conceptualization, Data curation, Formal analysis, Supervision, Funding acquisition, Methodology, Writing—review and editing; Vincenzo Esposito, Conceptualization, Resources, Methodology; Giovanni Bernardini, Conceptualization, Data curation, Supervision, Investigation, Writing—review and editing; Thomas Seyfried, Resources, Validation; Jakub Mieczkowski, Data curation, Software, Investigation, Methodology; Bozena Kaminska, Conceptualization, Data curation, Supervision, Writing—review and editing; Angela Santoni, Conceptualization, Supervision, Funding acquisition, Writing—review and editing; Cristina Limatola, Conceptualization, Supervision, Funding acquisition, Investigation, Writing—original draft, Project administration, Writing—review and editing

#### Author ORCIDs
Silvia Di Angelantonio (iD) http://orcid.org/0000-0003-1434-3648
Cristina Limatola (iD) http://orcid.org/0000-0001-7504-8197

#### Ethics
Animal experimentation: The protocol was approved by the Ministry of Health of Italy in accordance with the guidelines on the ethical use of animals from the EC council directive of September 22, 2010 (2010/63/EU).

#### Decision letter and Author response
Decision letter https://doi.org/10.7554/eLife.33415.021
Author response https://doi.org/10.7554/eLife.33415.022

## Additional files

#### Supplementary files
• Transparent reporting form
DOI: https://doi.org/10.7554/eLife.33415.019

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
