## [Decision Letter]

[Editors’ note: a previous version of this study was rejected after peer review, but the authors submitted for reconsideration. The first decision letter after peer review is shown below.]

Thank you for submitting your work entitled "Environmental stimuli shape microglial plasticity in glioma" for consideration by *eLife*. Your article has been reviewed by three peer reviewers, one of whom is a member of our Board of Reviewing Editors and the evaluation has been overseen by a Senior Editor. The reviewers have opted to remain anonymous.

Our decision has been reached after consultation between the reviewers. Based on these discussions and the individual reviews below, we regret to inform you that your work will not be considered further for publication in *eLife*.

Indeed, after careful consideration and discussion among the different reviewers and the Reviewing Editor, the conclusion was unanimous that this new study is interesting but requires a too substantial amount of additional data and analysis to support further consideration at this time.

Aside from concerns about the selection of the striatum for the work (which is not well justified), both reviewer #2 and reviewer #3 raised significant issues with the work as is. For instance, reviewer #2 was particularly concerned about the fact that published reports by others deny the central conclusion of your study, making it mandatory that additional cross validating and confirmatory experiments be performed (see appended review below for details). Likewise, reviewer #3 demands that you repeat the experiments using at least one additional mouse model (e.g. cell lines mentioned in the appended review are commercially available) on the basis that the GL261 cell line is so immunogenic that your entire paper may be the result of the brain's response to this single cell line.

Should you be able to fully address these major problems and all of the other more minor issues identified by the reviewers, we will be open to consider a substantial amended version of this manuscript. However, please note that even if fully revised, this re-submission will be subjected to a new full review process without any guarantee of publication.

Reviewer #1:

This is a quite interesting and original study even if the mechanistic link between EE and the shift in the microglial phenotype remains enigmatic. The main observations are that upon EE peri-GBM microglia and infiltrating cells adopt a pro-inflammatory and anti-tumor phenotype. This effect seems triggered by NK cells and mediated by IFgamma. The authors end their study by a series of investigations suggesting that EE operates via the production of BDNF.

While the study is quite elegant and provocative, the mains weakness for this reviewer is the molecular link between EE and the change in microglial phenotype, especially the experiments on BDNF that are poorly convincing and unclear.

Also, not clear if whether the selection of the striatum for the insertion of the GBM is important as EE was not reported to have even effects throughout the CNS.

EE thus far has been reported as stimulating neurotrophic factors and effects and thus how one goes from this to a pro-inflammatory phenotype remains unconvincing.

If the two sides of the brain are compared should a repeated measure design not used for the statistics?

Reviewer #2:

In the current study mice who were exposed to an enriched environment (EE) as defined as being housed 10 for cage and in the presence of an assortment of climbing ladders, seesaws, running wheel, balls, plastic, and wood objects and nesting material before brain implantation of human or murine glioma cells then left in such cages for additional 17 days revert an immunosuppressive phenotype of invading myeloid cells, by modulating inflammatory gene expression, increasing cell branching and patrolling activity, and promoting phagocytic activity. The modulations of gene expression were found to be dependent on interferon- (IFN) γ produced by NK cells and mimicked by interleukin-15 (IL-15). A role for BDNF linking environmental cues to the acquisition of a pro-inflammatory anti-tumor microenvironment in mouse brain was found.

The data presented are well illustrated and notably convincing. However, a singular time point analysis provided in this report coupled with notable conflicting data sets reported by others warrants a "real" need for rigor and the inclusion of cross-over analysis for this report to reach the muster for further consideration.

For example, while the reports cited (in Cell. 2010 Jul 9;142(1):52-64. doi: 10.1016/j.cell.2010.05.029 and PLoS One. 2012;7(12):e51525. doi: 10.1371/journal.pone.0051525. Epub 2012) demonstrated positive effects of EE on tumor regression in models of breast, melanoma and colon cancer subsequent research performed by others have failed to reproduce such findings (F1000Res. 2013 Jun 12;2:140. doi: 10.12688/f1000research.2-140.v1. eCollection 2013). While EE has shown effects in on neurogenesis, brain injury, cognitive capacity, memory, learning, neuronal vitality, Alzheimer's and Parkinson's disease, anxiety, depression, drug addiction, amongst others the impact levels have not been consistent throughout (recent examples reported in Behav Brain Res. 2016 Mar 15;301:72-83. doi: 10.1016/j.bbr.2015.12.028; Behav Brain Res. 2017 Jun 16;332:355-361. doi: 10.1016/j.bbr.2017.06.009. [Epub ahead of print]. This has led others to question differences based on differential tumor growth rates, microbiota of the mice housed in different facilities, variations in noise and handling between two environments as well as food or cage composition. As EE may not be consistent for results regarding decreased tumor growth this puts forward a need to provide further controls for the possibility of other influences that factors could affect cancer progression and related inflammatory profiles in this setting. Having such additional data in hand would increase the impact substantively of this important study.

On balance, I am certainly open to the fact that different laboratories may see different results based on investigator preferences and laboratory practice it is incumbent on the Journal to request as is responsible and reasonable for the authors to engage in cross validations of their data sets. Herein, placing SE housed mice into EE environments before and during tumor implantation and vice versa. Analyses of the animal microbioata and blood/plasma biomarkers during the crossed experiments would also prove helpful in validating the current data sets.

Reviewer #3:

The authors have chosen to study an interesting and potentially clinically relevant aspect of glioblastioma biology, that of microglial/macrophage interactions. This group has elected to use implanted GL261 cells in syngeneic, immunocompetent mice as the primary model for these studies. This group's previous work demonstrated that an enriched housing environment (EE) reduces glioma cell growth. In this report, the authors provide needed mechanistic details to explain this observation. By in large, this is a well-designed series of experiments that is of interest to both the glioma and microglia/macrophage communities. I do have a few concerns that should be addressed prior to publication:

1) The near-exclusive use of a single model (GL261) limits the interpretation of these provocative results. This cell line is known to highly express MHCI, and is in and of itself moderately immunogeneic. It is therefore recommended that the authors repeat key figures (Figure 4 and Figure 5) with at least one additional syngeneic model. GL26 and CT-2A are cell lines syngeneic in C57Bl/6, and would be relatively straightforward in this regard.

2) The authors did make use of the human cell line U87 in a xenograft system. However, based on the text, figures and figure legends, it is often unclear when this model was used. This is most confusing in Figure 5—figure supplement 1. In panels B–E, it is unclear which model was employed for which experiment. Does EE/SE imply only C57Bl/6? Please be explicit.

3) Figure 4 schematic and legend should be edited to make it clear that the CD11b(+) cells were isolated and assayed from the GBM explants- and not all cells. CD11b(+) cells isolated from nearly any organ in nearly any disease or quiescent state, once treated with IFN-γ ex vivo, are expected to express pro-inflammatory genes. This panel could be eliminated from the manuscript completely.

4) In general, the clinical implications of the results should be toned down considerably. There are nearly no correlative studies presented that make productive use of human tissues or clinical data. Response of a single, immunogenic model to treatment with IL15 treatment, while provocative, is not nearly sufficient to justify human studies.

[Editors’ note: what now follows is the decision letter after the authors submitted for further consideration.]

Thank you for resubmitting your work entitled "Environmental stimuli shape microglial plasticity in glioma" for further consideration at *eLife*. Your revised article has been favorably evaluated by Michel Nussenzweig (Senior editor), a Reviewing editor, and two reviewers.

The manuscript has been improved but there are some remaining issues that need to be addressed before acceptance, as outlined below:

As stressed by reviewer #1, it would be important for the readers to have the authors discuss more thoroughly the opposite results reported in the published literature and the possible reasons for these divergent results. Moreover, the authors are encouraged to have the manuscript edited for English to avoid potential confusions and misunderstanding and maximize impact.

*Reviewer #1:*

The authors have responded to my queries and included appropriate cross validation data sets. However, since there are clear opposite results reported in the published literature I would have expected to see some level of discussion of the conflicting data sets with opinions as to why they might have occurred. This I believe important to the field and should be addressed prior to publication.

Reviewer #2:

Since last submission, the authors have addressed our major concerns. Namely:

1) They have added one additional model (CT-2a) to supplement the workhorse model of the paper (GL261), and repeated key experiments in this system.

2) The figures and figure legends have been clarified. In particular, the changes in Figure 5—figure supplement 1 and Figure 4 are appreciated.

3) The discussion and conclusions have been tightened up to better correlate with the data presented.

The new manuscript provides needed mechanism to further explain the group's previous work that an enriched housing environment (EE) reduces glioma cell growth in animal models.

Prior to publication, the manuscript will require some additional editing; some grammatical and English usage errors remain.

---

## [Author Response]

[Editors’ note: the author responses to the first round of peer review follow.]

[…] Aside from concerns about the selection of the striatum for the work (which is not well justified), both reviewer #2 and reviewer #3 raised significant issues with the work as is. For instance, reviewer #2 was particularly concerned about the fact that published reports by others deny the central conclusion of your study, making it mandatory that additional cross validating and confirmatory experiments be performed (see appended review below for details). Likewise, reviewer #3 demands that you repeat the experiments using at least one additional mouse model (e.g. cell lines mentioned in the appended review are commercially available) on the basis that the GL261 cell line is so immunogenic that your entire paper may be the result of the brain's response to this single cell line.Should you be able to fully address these major problems and all of the other more minor issues identified by the reviewers, we will be open to consider a substantial amended version of this manuscript. However, please note that even if fully revised, this re-submission will be subjected to a new full review process without any guarantee of publication.

1) Concerns about the selection of the striatum for the work (which is not well justified):

We decided to inject the tumor cells in the striatal region for the following reasons:

i) It is a widely used protocol, validated by high number of studies, that permits reproducibility of tumor cell migration, vascularization and monitoring of the effect of different treatments on tumor size and mice survival (Renner et al., 2015; Clark et al., 2014; Lenting et al., 2017).

ii) The choice of different regions of injection (such as some area of the cerebral cortex), could induce rapid neurologic deficits in mice, not necessary for our study. Injecting glioma in the striatal region (apparently) avoids this early stress, which is important for ethical reasons and for our experimental settings.

iii) Not least, one established hypothesis is that glioma cells derive from modified stem cells (Singh et al., 2003; Galli et al., 2004), which are abundant in the subventricular region (Nunes et al., 2003; Smith et al., 2016). Normal or already transformed cells, originating from neurogenic niches, would migrate in brain parenchyma and transform at an unknown moment during their invasion. We reasoned that injecting tumor cells deep in the striatum, not far from the SVZ, would create a more realistic scenario of cell movement and growth in brain parenchyma, sensing and responding to the regional expression of specific extracellular matrix proteins.

2) Published reports by others deny the central conclusion of your study, making it mandatory that additional cross validating and confirmatory experiments be performed.

We are aware that contrasting results on the effects of environment on tumors have been published by different laboratories. It is a remarkable aspect of these studies that, we are convinced, deserves further investigation to understand which could be the confounding factors responsible of these differences.

To answer the reviewer, we performed additional experiments. In detail:

i) We repeated some of the experiments (effects of SE and EE on tumor size) in a different room, and the results do not change (now mentioned in the Materials and methods section and Results section).

ii) Some experiments (effects of SE and EE on tumor size) were performed by a different researcher, and the results do not change (now mentioned in the Materials and methods and Results sections, see above).

iii) The effect of animal handling was investigated in SE housed mice. At this aim, mice in SE were manipulated for the same time dedicated to EE-housed animals during toys change and cage modification. Results do not change, and are not shown in the text (SE: 4.19 ± 0.60 mm3 n=10; SE manipulated: 3.82 ± 0.14 mm3 n=5) but mentioned in the Materials and methods and Results sections (see above).

iv) A group of mice was first housed in EE and after tumor injection, in SE. Results demonstrate that this treatment was as effective as EE-tumor injection-EE, but these data are only mentioned in the method and result sections, see above (SE: 4.19 ± 0.6 mm3; EE: 0.5 ± 0.14 mm3 n = 6 one Way ANOVA p<0.001 Dunn’s methods vs SE). Please note that the opposite experiment was already done in Garofalo et al., 2015. In that paper we housed mice in SE, injected the tumor, and moved mice in EE: this treatment did not result in a significant reduction of tumor size, demonstrating that the period of enrichment must precede tumor presence.

v) In addition, considering the importance to understand the possible reasons of contrasting results among different laboratories, and to better compare protocols, we decided to insert a table (new Table 1), describing all the details of our housing conditions, comprising the sanitary report, as suggested by Westwood, (2013).

3) Repeat the experiments using at least one additional mouse model (e.g. cell lines mentioned in the appended review are commercially available) on the basis that the GL261 cell line is so immunogenic that your entire paper may be the result of the brain's response to this single cell line.

We thank the reviewer for this important consideration. We repeated some of the experiments with one of the suggested mouse cell line, CT-2a. Qualitatively, the results do not differ from what observed with GL261; we only observed some differences in the extent of reduction in tumor volume and in the modulation of gene expression in CD11b+ cells. There is only one significant difference in the response to the treatment with anti-IFNγ, where the effect of EE on tumor size is lost (new Figure 4—figure supplement 2) while with GL261 we observed a partial reduction (Figure 4). The new results are now shown in the Results section and in the figure supplements. Specifically:

i) Exposing mice to an EE was effective in reducing tumor size also in mice inEected with CT-2a (SE: 19.43 ± 3.4 mm3; EE: 8.58 ± 2.4 mm3, n=5 p<0.001 one Way ANOVA p<0.001 Holm Sidak). Figure 1—figure supplement 1.

ii) We also isolated CD11b positive cells from these mice and the results obtained show that in mice injected with CT-2a and exposed to EE, there is a significant increase of expression of pro-inflammatory genes and a reduction of anti-inflammatory ones. Figure 1—figure supplement 1), in accordance with what observed in CD11b+ cells from mice bearing GL261 tumor (Figure 1).

iii) Mice injected with CT-2a were treted with Ab-IFNγ: under these conditions, no differences were induced by the housing conditions. SE: 27.26 ± 1.32 mm3; EE: 27.36 ± 1.35 n=4 (one way ANOVA p<0.001 Holm Sidak vs EE), see new Figure 4—figure supplement 2, right). CD11b+ cells isolated from these mice had no differences in the expression level of pro- or anti-inflammatory genes in EE vs SE (new Figure 4—figure supplement 2).

iv) CT2a-injected mice were implanted with osmotic pumps releasing IL-15, as described before, and a significant reduction of tumor size was observed (Veh 14.70 ± 1.20 mm3 n= 4; IL-15: 5.32 ± 0.89 mm3 n=4 t test p=0.003) (Figure 5—figure supplement 1). CD11b+ cells isolated from these mice upregulated pro-inflammatory and down-regulated anti-inflammatory genes upon IL-15 treatment (new Figure 5—figure supplement 1).

Reviewer #1:This is a quite interesting and original study even if the mechanistic link between EE and the shift in the microglial phenotype remains enigmatic. The main observations are that upon EE peri-GBM microglia and infiltrating cells adopt a pro-inflammatory and anti-tumor phenotype. This effect seems triggered by NK cells and mediated by IFgamma. The authors end their study by a series of investigations suggesting that EE operates via the production of BDNF.While the study is quite elegant and provocative, the mains weakness for this reviewer is the molecular link between EE and the change in microglial phenotype, especially the experiments on BDNF that are poorly convincing and unclear.

We thank the reviewer for this comment. We investigated the molecular link between EE and the induction of microglial phenotype: we have previously shown that EE induces an increase of brain BDNF (Garofalo et al., 2015); data shown in this paper demonstrate that BDNF infusion in the brain of glioma bearing mice induces IL-15 transcription in CD11b+ cells (shown in Figure 5). IL-15 infusion in glioma bearing mice, in turn, switches microglia phenotype toward a pro-inflammatory phenotype (Figure 5). We believe that this is an acceptable explanation of the molecular link between EE and the modulation of microglia phenotype.

In addition, further experiments performed in *Bdnf^+/-^*mice, injected with glioma, demonstrated that EE was not able to induce an increase of IL-15 transcription when the level of BDNF was low, thus demonstrating the link between EE and IL-15, through BDNF (Figure 5).

Furthermore, in vitro experiments on primary microglia demonstrate that BDNF stimulation induces an increase of IL-15 transcription only when co-cultured with glioma (or stimulated with IL-4), Figure 5.

All together these experiments provide evidence that in the tumor microenvironment there is a link between BDNF and the acquisition of a pro-inflammatory (protective) microglia phenotype. Figure 5 shows that this effect is not observed in unstimulated or LPS/IFNγ treated microglia, but requires a tumor microenvironment. So, we believe that this result is in line with the neurotrophic/protective effect of this neurotrophin.

Also, not clear if whether the selection of the striatum for the insertion of the GBM is important as EE was not reported to have even effects throughout the CNS.EE thus far has been reported as stimulating neurotrophic factors and effects and thus how one goes from this to a pro-inflammatory phenotype remains unconvincing.If the two sides of the brain are compared should a repeated measure design not used for the statistics?

We decided to inject the tumor cells in the striatal region for the following reasons:

i) It is a widely used protocol, validated by high number of studies, that permits reproducibility of tumor cell migration, vascularization and monitoring of the effect of different treatments on tumor size and mice survival (Renner et al., 2015; Clark et al., 2014; Lenting et al., 2017);

ii) The choice of different regions of injection (such as some area of the cerebral cortex), could induce rapid neurologic deficits in mice, not necessary for our study. Injecting glioma in the striatal region (apparently) avoids this early stress, which is important for ethical reasons and for our experimental settings.

iii) Not least, one established hypothesis is that glioma cells derive from modified stem cells (Singh et al., 2003; Galli et al., 2004), which are abundant in the subventricular region (Nunes et al., 2003; Smith et al., 2016). Normal or already transformed cells, originating from neurogenic niches, would migrate in brain parenchyma and transform in an unknown moment during their invasion. We reasoned that injecting tumor cells deep in the striatum, not far from the SVZ, would create a more realistic scenario of cell movement and growth in brain parenchyma, sensing and responding to the regional expression of specific extracellular matrix proteins.

We agree with the reviewer that the effects of EE are largely region specific. However, we provide evidence that upon EE there are modifications also in the striatal (CLH) regions, in microglia, for voltage dependent potassium currents (Figure 2) and process movement (Figure 2). This would indicate that soluble factors locally produced could exert their effects also in the striatum.

Reviewer #2:[…] On balance, I am certainly open to the fact that different laboratories may see different results based on investigator preferences and laboratory practice it is incumbent on the Journal to request as is responsible and reasonable for the authors to engage in cross validations of their data sets. Herein, placing SE housed mice into EE environments before and during tumor implantation and vice versa. Analyses of the animal microbioata and blood/plasma biomarkers during the crossed experiments would also prove helpful in validating the current data sets.

We are aware that contrasting results on the effects of environment on tumors have been published by different laboratories. It is a remarkable aspect of these studies that, we are convinced, deserves further investigation to understand which could be the confounding factors responsible of these differences.

To answer the reviewer, we performed additional experiments. In detail:

i) We repeated some of the experiments (effects of SE and EE on tumor size) in a different room, and the results do not change (now mentioned in the Materials and methods section; and Results section);

ii) Some experiments (effects of SE and EE on tumor size) were performed by a different researcher, and the results do not change (now mentioned in the Materials and methods and Results sections, see above);

iii) The effect of animal handling was investigated in SE housed mice. At this aim, mice in SE were manipulated for the same time dedicated to EE-housed animals during toys change and cage modification. Results do not change, and are not shown in the text (SE: 4.19 ± 0.60 mm3 n=10; SE manipulated: 3.82 ± 0.14 mm3 n=5) but mentioned in the methods and results (see above).

iv) A group of mice was first housed in EE and after tumor injection, in SE. Results demonstrate that this treatment was as effective as EE-tumor injection-EE, but these data are only mentioned in the method and result sections, see above (SE: 4.19 ± 0.6 mm3; EE: 0.5 ± 0.14 mm3 n = 6 one Way ANOVA p<0.001 Dunn’s methods vs SE). Please note that the opposite experiment was already done in Garofalo et al., 2015. In that paper we housed mice in SE, injected the tumor, and moved mice in EE: this treatment did not result in a significant reduction of tumor size, demonstrating that the period of enrichment must precede tumor presence.

v) In addition, considering the importance to understand the possible reasons of contrasting results among different laboratories, and to better compare protocols, we decided to insert a table (new Table 1), describing all the details of our housing conditions, comprising the sanitary report, as suggested by Westwood, (2013).

In our previous paper (Garofalo et al., 2015) we analyzed tumor size at two different time points (10 and 17 days). We did not repeat this analysis for CD11b+ cell phenotype but decided to focus on a single time.

Reviewer #3:The authors have chosen to study an interesting and potentially clinically relevant aspect of glioblastioma biology, that of microglial/macrophage interactions. This group has elected to use implanted GL261 cells in syngeneic, immunocompetent mice as the primary model for these studies. This group's previous work demonstrated that an enriched housing environment (EE) reduces glioma cell growth. In this report, the authors provide needed mechanistic details to explain this observation. By in large, this is a well-designed series of experiments that is of interest to both the glioma and microglia/macrophage communities. I do have a few concerns that should be addressed prior to publication:1) The near-exclusive use of a single model (GL261) limits the interpretation of these provocative results. This cell line is known to highly express MHCI, and is in and of itself moderately immunogeneic. It is therefore recommended that the authors repeat key figures (Figure 4 and Figure 5) with at least one additional syngeneic model. GL26 and CT-2A are cell lines syngeneic in C57Bl/6, and would be relatively straightforward in this regard.

We thank the reviewer for this important consideration. We repeated some of the experiments with one of the suggested mouse cell line, CT-2a. Qualitatively, the results do not differ from what observed with GL261; we only observed some differences in the extent of reduction in tumor volume and in the modulation of gene expression in CD11b+ cells. There is only one significant difference in the response to the treatment with anti-IFNγ, where the effect of EE on tumor size is lost (New Figure 4—figure supplement 2) while with GL261 we observed a partial reduction (Figure 4). The new results are now shown in part in the Result section, and in part as Supplement to Figures. In detail:

i) Exposing mice to an EE was effective in reducing tumor size also in mice injected with CT-2a (SE: 19.43 ± 3.4 mm3; EE: 8.58 ± 2.4 mm3, n=5 p<0.001 one Way ANOVA p<0.001 Holm Sidak). Figure 1—figure supplement 1.

ii) We also isolated CD11b positive cells from these mice and the results obtained show that in mice injected with CT-2a and exposed to EE, there is a significant increase of expression of pro-inflammatory genes and a reduction of anti-inflammatory ones. Figure 1—figure supplement 1), in accordance with what observed in CD11b+ cells from mice bearing GL261 tumor (Figure 1).

iii) Mice injected with CT-2a were treted with Ab-IFNγ: under these conditions, no differences were induced by the housing conditions. SE: 27.26 ± 1.32 mm3; EE: 27.36 ± 1.35 n=4 (one way ANOVA p<0.001 Holm Sidak vs EE), see new Figure 4—figure supplement 2, right). CD11b+ cells isolated from these mice had no differences in the expression level of pro- or anti-inflammatory genes in EE vs SE (new Figure 4—figure supplement 2).

iv) CT2a-injected mice were implanted with osmotic pumps releasing IL-15, as described before, and a significant reduction of tumor size was observed (Veh 14.70 ± 1.20 mm3 n= 4; IL-15: 5.32 ± 0.89 mm3 n=4 t test p=0.003) (Figure 5—figure supplement 1). CD11b+ cells isolated from these mice upregulated pro-inflammatory and down-regulated anti-inflammatory genes upon IL-15 treatment (new Figure 5—figure supplement 1).

2) The authors did make use of the human cell line U87 in a xenograft system. However, based on the text, figures and figure legends, it is often unclear when this model was used. This is most confusing in Figure 5—figure supplement 1. In panels B–E, it is unclear which model was employed for which experiment. Does EE/SE imply only C57Bl/6? Please be explicit.

We are sorry for this lack of clarity. We now label the Figure 5—figure supplement 1, to be as explicit as possible. The effect of EE on U87-MG xenograft on tumor size, NK cells infiltration and response to IL-15 was already shown in our previous paper (Garofalo et al., 2015). For this reason the effect of IL-15 on infiltrating NK cells in detail is shown here only for GL261.

3) Figure 4 schematic and legend should be edited to make it clear that the CD11b(+) cells were isolated and assayed from the GBM explants- and not all cells. CD11b(+) cells isolated from nearly any organ in nearly any disease or quiescent state, once treated with IFN-γ ex vivo, are expected to express pro-inflammatory genes. This panel could be eliminated from the manuscript completely.

We are sorry for the misunderstanding. We did stimulate a mixture of cells obtained from dissociated tissue (CD11b+ cells, tumor cells, and other cells). This si now better explained in the figure legend. This implicate that the response cannot be anticipated or compared with that obtained with IFNγ-treated pure CD11b+ cells.

Even if expected, in general, we believe that these data, obtained on human tissue/cells, provide additional value to our results. These are the only experiments we report on human tissue and validate our experimental hypothesis.

4) In general, the clinical implications of the results should be toned down considerably. There are nearly no correlative studies presented that make productive use of human tissues or clinical data. Response of a single, immunogenic model to treatment with IL15 treatment, while provocative, is not nearly sufficient to justify human studies.

We are sorry but we cannot answer the review on this point, because we cannot find the text about the clinical implications of our study.

[Editors' note: the author responses to the re-review follow.]

Reviewer #1:The authors have responded to my queries and included appropriate cross validation data sets. However, since there are clear opposite results reported in the published literature I would have expected to see some level of discussion of the conflicting data sets with opinions as to why they might have occurred. This I believe important to the field and should be addressed prior to publication.

We now added another comment in the Discussion section that suggests possible explanations for the different results published on the effects of environment on tumors. We also added a new reference – Goldszmid et al., (2015).

“Contrasting studies described either positive or no effects on tumor progression for EE, considered as a sum of eu-stressing stimuli (13, 16) The reason of these differences is not clear and additional experimental approaches such as analyses of the animal microbiota could be useful. It is indeed possible that central and peripheral signals activated in EE housed mice could modulate microbial composition of the gut, in a bidirectional communication with the different tumor microenvironment (42).”

2) Prior to publication, the manuscript will require some additional editing; some grammatical and English usage errors remain.

Careful re-editing has been performed.